# Solar-driven upgrading of biomass by coupled hydrogenation using in situ (photo)electrochemically generated H₂

Keisuke Obata [1,2], Michael Schwarze[3], Tabea A. Thiel[3,4], Xinyi Zhang[1], Babu Radhakrishnan[1], Ibbi Y. Ahmet [1], Roel van de Krol [1,3], Reinhard Schomäcker [3] & Fatwa F. Abdi [1,5]

With the increasing pressure to decarbonize our society, green hydrogen has been identified as a key element in a future fossil fuel-free energy infrastructure. Solar water splitting through photoelectrochemical approaches is an elegant way to produce green hydrogen, but for low-value products like hydrogen, photoelectrochemical production pathways are difficult to be made economically competitive. A possible solution is to co-produce value-added chemicals. Here, we propose and demonstrate the in situ use of (photo)electrochemically generated H₂ for the homogeneous hydrogenation of itaconic acid—a biomass-derived feedstock—to methyl succinic acid. Coupling these two processes offers major advantages in terms of stability and reaction flexibility compared to direct electrochemical hydrogenation, while minimizing the overpotential. An overall conversion of up to ~60% of the produced hydrogen is demonstrated for our coupled process, and a techno-economic assessment of our proposed device further reveals the benefit of coupling solar hydrogen production to a chemical transformation.

The last couple of decades have witnessed increasing utilization of solar energy to overcome the scarcity of fossil fuels and the growing concerns about global warming. Due to the intermittency of sunlight, green hydrogen production via photovoltaic-powered electrolysis of water is considered a viable long-term energy storage option. One disadvantage of this approach is the lack of thermal coupling between the photovoltaic panels (which can reach 60–80 °C, decreasing their efficiency) with the electrolysis process (which would benefit from solar heat). In the photoelectrochemical (PEC) approach, efficient thermal coupling is ensured by combining light absorption and electrochemistry within a single device. In a PEC device, the electrons and holes generated in the light-absorbing semiconductors drive the hydrogen and oxygen evolution reactions (HER and OER, respectively) directly at the surface of the semiconductor. Solar-to-hydrogen

efficiencies ($\eta_{STH}$) well above 10% have been demonstrated[1,2], but several techno-economic assessments revealed that the levelized cost of hydrogen (LCOH) is still far from being competitive (~10 USD/kg vs. ~1.4 USD/kg for hydrogen from steam methane reforming)[3]. One way to increase competitiveness is to directly use the hydrogen, or part of it, to upgrade biomass feedstock. Although the total market size of hydrogen is much larger than that of any single biomass upgrading product, solar-driven biomass upgrading offers an alternative and green pathway to fossil fuel-based chemical production processes. For example, instead of the OER, (photo)electrochemical oxidation of 5-hydroxymethylfurfural to 2,5-furan dicarboxylic acid has been considered as an alternative anodic reaction[4]. Similarly, on the cathode, biomass feedstock containing oxygen and unsaturated carbon bonds can be (photo)electrochemically hydrogenated to valuable

[1]Institute for Solar Fuels, Helmholtz-Zentrum Berlin für Materialien und Energie GmbH, Hahn-Meitner-Platz 1, 14109 Berlin, Germany. [2]Department of Chemical System Engineering, School of Engineering, The University of Tokyo, Tokyo 113-8656, Japan. [3]Technische Universität Berlin, Department of Chemistry, Straße des 17. Juni 124, 10623 Berlin, Germany. [4]Leibniz Institute for Catalysis, Albert-Einstein-Straße 29a, 18059 Rostock, Germany. [5]School of Energy and Environment, City University of Hong Kong, 83 Tat Chee Avenue, Kowloon, Hong Kong SAR, China. ✉e-mail: ffabdi@cityu.edu.hk

chemicals[5,6]. However, the cathodic hydrogenation reactions often suffer from competing HER and poor stability[7,8].

In this study, we demonstrate an integrated solar-driven device in which the photoelectrochemical production of hydrogen is coupled to a catalytic hydrogenation reaction. A Rh-based homogeneous hydrogenation catalyst is introduced into the catholyte solution to overcome the selectivity and stability issues of direct (i.e., decoupled) electrochemical hydrogenation reactions. As shown in Fig. 1, (photo) electrochemically generated hydrogen is used in situ for the subsequent coupled homogeneous hydrogenation reaction. By designing the homogeneous catalyst structure, the reactivity, and selectivity towards various hydrogenation reactions can be tuned[9]. As a model reaction, we couple (photo)electrochemical hydrogen production with the hydrogenation of itaconic acid (IA) to methyl succinic acid (MSA). IA is one of the major sugar-derived building blocks while MSA is a solvent or a feed in cosmetics and polymer manufacturing with an estimated global market size of up to ~15,000 t[10–13]. Moreover, homogeneous hydrogenation of IA to MSA is known to occur at room temperature and 1 bar of $H_2$[9,14]. Therefore, this reaction is suitable as a proof-of-concept of our coupled approach and to clarify its advantages as compared with the conventional pathway. Our coupled approach achieves continuous hydrogenation with up to 60% $H_2$-to-MSA conversion. Importantly, no deactivation was observed, in contrast to the direct hydrogenation approach. A techno-economic assessment shows that such selectivity allows the coupled photoelectrochemical hydrogenation device to be economically profitable and generate hydrogen at a cost that is competitive with steam methane reforming. Finally, the potential, limitations, and future direction of our present concept are further discussed.

## Results and discussion
### Direct vs. coupled electrochemical hydrogenation of IA
Homogeneous hydrogenation of IA at ambient conditions is initially investigated with a Rh/TPPTS catalyst in a semi-batch reactor (Fig. S1 and note S1). The mechanism of homogeneous hydrogenation using a Rh/TPPTS catalyst has been reported elsewhere[15–17]. The obtained optimum IA and catalyst concentrations are then used for the dark electrochemical hydrogenation of IA in an electrochemical flow cell with a Pt cathode and a Dimensionally Stable Anode (DSA®) as the counter electrode (Fig. S2). The uncompensated resistances of the electrochemical cells in 1 M $KP_i$ using three- and two-electrode

configurations were 0.5 and 1.6 Ω, respectively. Two different approaches are considered: direct and coupled electrochemical hydrogenation. For the direct approach, only IA is added to the 1 M $KP_i$ catholyte. For the coupled approach, the catholyte also contains the Rh/TPPTS catalyst. A comparison between these approaches allows us to investigate the benefits, if any, of coupling the electrochemical hydrogen production and the hydrogenation reaction.

We first discuss the direct electrochemical hydrogenation of IA on a Pt electrode. The adsorption of IA and Rh/TPPTS on Pt was investigated using underpotential deposition cyclic voltammetry (see Fig. S3 and note S2). Linear sweep voltammograms (LSV) in 1 M $KP_i$ (black) and 1 M $KP_i$ + 0.15 M IA (red) are shown in Fig. 2a. In the presence of IA, an additional cathodic shoulder is observed in the range from 0 to −0.05 V vs. reversible hydrogen electrode (RHE). This potentially indicates a direct heterogeneous electrochemical reduction of IA on the Pt surface. Indeed, electrochemical hydrogenation of C=C bonds in maleic acid and unsaturated fatty acids has been reported on Pt/C electrodes[18,19]. However, a continuous potential decrease is observed in the chronopotentiometry data (Fig. 2b), reaching a more negative potential than in pure $KP_i$. After 90 min at −2 mA cm⁻², the cathodic shoulder has disappeared (Fig. S4), suggesting that HER becomes the dominant reaction. This has indeed been confirmed by ¹H-NMR product quantification (Fig. 2b). Although the hydrogenation product, MSA, can be clearly detected, the production rate continuously decreases, suggesting that the heterogeneous hydrogenation process is unstable (Figs. 2b–red and S5a). The hydrogenation terminates after ~120 min. A similar deactivation has been reported during the electrochemical hydrogenation of maleic acid on Pt/C[19], which may indicate a general limitation for heterogeneous electrochemical hydrogenation reactions at Pt surfaces. We attribute the instability to Pt surface transformations during hydrogenation. Analysis of the peaks in the voltammogram of Pt-H underpotential deposition reveals that IA adsorbs on Pt (110) and (100) facets (Fig. S3 and note S2). These facets were reported to undergo reconstruction, anion accumulation, and deposition of impurities under prolonged cathodic bias even in the absence of organic substrates[20], all of which may lead to the loss of active sites and the decay of the heterogeneous hydrogenation rate.

We now turn our attention to the coupled electrochemical hydrogenation, in which the hydrogen that evolves on the Pt electrode is used for the catalytic hydrogenation of IA over a Rh/TPPTS catalyst within the same catholyte compartment. The addition of the Rh/TPPTS

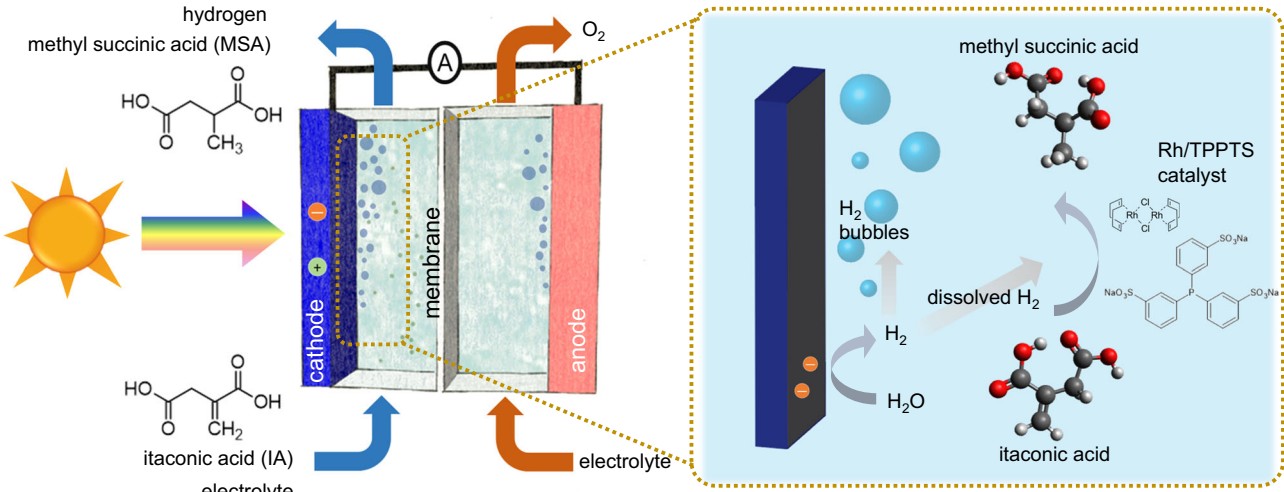

**Fig. 1 | Scheme of our proposed coupled (photo)electrochemical hydrogenation system.** The catholyte and anolyte cells are separated by a cation exchange membrane, and (photo)electrochemically generated $H_2$ is (partially) utilized in situ to perform hydrogenation reactions (in this case itaconic acid to methyl succinic acid with a homogeneous Rh/TPPTS catalyst) within the catholyte chamber.

**a**

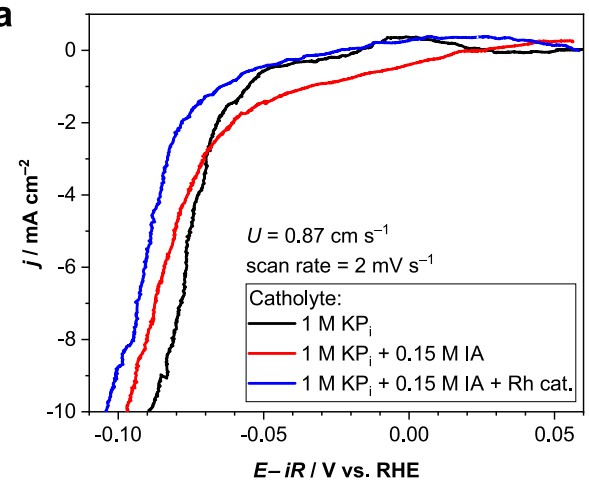

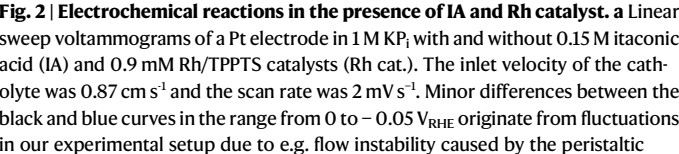

**b**

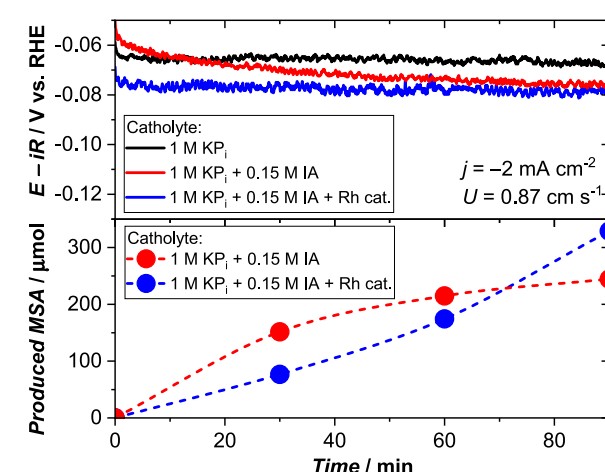

**Fig. 2 | Electrochemical reactions in the presence of IA and Rh catalyst. a** Linear sweep voltammograms of a Pt electrode in 1 M $KP_i$ with and without 0.15 M itaconic acid (IA) and 0.9 mM Rh/TPPTS catalysts (Rh cat.). The inlet velocity of the catholyte was 0.87 cm s⁻¹ and the scan rate was 2 mV s⁻¹. Minor differences between the black and blue curves in the range from 0 to − 0.05 $V_{RHE}$ originate from fluctuations in our experimental setup due to e.g. flow instability caused by the peristaltic pump, as also shown in Fig. S13a. **b** Chronopotentiometry curves (top) and the cumulative amount of produced MSA determined from ¹H-NMR measurements (bottom) at a current density of −2 mA cm⁻² in 1 M $KP_i$ with IA or IA + Rh cat. For all measurements, the anode was DSA® and the anolyte was 1 M $KP_i$. The uncompensated resistance estimated from an impedance measurement was approximately 0.5 Ω.

catalyst results in the disappearance of the cathodic shoulder in the range between 0 and −0.05 V vs. RHE (Fig. 2a–blue), which is present in the LSV curve measured without the Rh/TPPTS catalyst (Fig. 2a–red). This suggests that the presence of Rh/TPPTS in the electrolyte prevents the direct electrochemical IA hydrogenation on Pt, most likely due to the blockage of the active sites by Rh/TPPTS catalysts and excess of TPPTS ligands. Chronopotentiometry data (Fig. 2b top) shows that the potential remains constant, albeit slightly more negative than in $KP_i$; we attribute this to the partial blocking of Pt active sites due to the adsorption of IA (vide supra). Interestingly, although the cathodic shoulder that we attributed to the occurrence of direct heterogeneous IA hydrogenation (between 0 and −0.05 V vs. RHE) disappears, MSA is clearly detected from the product analysis (Fig. 2b bottom–blue). This can be explained by homogeneous IA hydrogenation over the Rh/TPPTS catalyst, using the hydrogen that was generated in situ at the Pt surface. To the best of our knowledge, this is the first demonstration of coupled hydrogenation using homogeneous catalysts and in situ generated H₂. More importantly, in contrast to the direct electrochemical hydrogenation, the total amount of produced MSA continues to increase without any sign of deactivation in the presence of IA and Rh catalysts (Figs. 2b and S5b). This indicates a significant advantage of the coupled approach over the direct electrochemical hydrogenation approach. At −2 mA cm⁻², the H₂-to-MSA conversion was 60 ± 18%, whilst ~40% of the H₂ remains unconverted. Control measurements with various additives suggest that adding the Rh catalyst is key to preventing deactivation associated with the presence of IA in the catholyte (Fig. S6a), despite the slight increase of overpotential (Fig. S6b). The Rh catalyst is also stable within our experimental conditions using the Pt electrode (Fig. S7). The mechanism by which the Rh catalyst prevents the deactivation of IA hydrogenation is not yet fully understood and is beyond the scope of this work. Although the demonstrated total production of MSA using our coupled electrochemical hydrogenation was limited to 800 μmol in the present test (Fig. S5b), further production should be possible without any noticeable deactivation until the IA concentration fully depletes in the solution. Indeed, our experiments in a semi-batch reactor using an external feed of H₂ (Fig. S1) show no deactivation of the hydrogenation rate up until the IA is fully depleted. The total produced MSA in this case was estimated to be 7.8 mmol. Further

recyclability tests of the Rh/TPPTS catalysts have been discussed in our previous reports[9,21].

We briefly note that the initial production of MSA (first ~60 min in Fig. 2b) was lower in the presence of Rh/TPPTS catalysts (blue) than without Rh/TPPTS catalysts (red). This may at first seem surprising, since both heterogeneous and homogeneous hydrogenation can, in principle, occur in the presence of Rh/TPPTS. However, as shown by the disappearance of the cathodic shoulder in Fig. 2a, the heterogeneous pathway on Pt is suppressed when both IA and the Rh/TPPTS catalyst are present. The homogeneous pathway is therefore the dominant one, even though its initial kinetics are slower than those of the heterogeneous pathway. Moreover, the coupled hydrogenation approach with only homogeneous hydrogenation shows excellent stability, while the production rate for the heterogeneous pathway (i.e., the direct electrochemical approach) decreases to a point that the reaction completely terminates after ~120 min (Fig. S5a).

Direct electrochemical IA hydrogenation to MSA has been demonstrated before by Holzhäuser et al.[12]. To the best of our knowledge, this is the only other report currently available on electrochemical hydrogenation of IA to MSA. In that study, they investigated various metal electrodes inactive for HER in acidic solutions (i.e., Pb, Cu, Cu-Pb, Ni, Fe) at several applied potentials and found that Pb shows the highest activity for IA hydrogenation. Initial faradaic efficiencies (first 10 mins) as high as ~60% were reported, but the long-term stability of the reaction was not reported since a batch reactor was used instead of a flow reactor. Nonetheless, even if the reaction were stable, the onset potential for the reaction was −1 V vs RHE, indicating a significant overpotential. In contrast, since our concept relies on electrochemically produced hydrogen, the overpotential can be minimized—our onset potential lies at ~0 V vs. RHE—while achieving a similarly high H₂-to-MSA conversion of 60 ± 18%. The MSA yield of direct hydrogenation using Pb might be improved by adding Rh/TPPTS catalyst; Rh/TPPTS could utilize the H₂ generated as a byproduct at the Pb electrode to hydrogenate IA to MSA. However, considering the huge overpotential on Pb and that the stability of Rh/TPPTS within the potential range is unknown, we keep this concept of combining direct and coupled hydrogenation in a single device for future investigations. We also note that the use of Pt as a model cathode in this study can be replaced with alternative earth-abundant

HER catalysts such as NiMo, transition metal dichalcogenides, and phosphides[22–24].

To further explore the advantages offered by the coupled approach, the influence of various parameters on the resulting $H_2$-to-MSA conversion is investigated. The electrolyte flow is expected to affect the concentration overpotentials, bubble formation, and $H_2$ retention time in the electrolyte[25,26]. Experiments performed at different flow rates reveal that the overpotential remains relatively constant (Fig. S8 and note S3) and the $H_2$-to-MSA conversion increases with increasing catholyte inlet velocity (Figs. 3a and S5b). The latter is surprising at first since it contradicts our numerical simulation results (Fig. 3a, see note S4 for model description). Using the homogeneous kinetic parameters obtained in Fig. S1, our simulations predict that the $H_2$-to-MSA conversion at higher electrolyte velocity would decrease due to the reduced $H_2$ retention time in the electrolyte. We attribute this discrepancy to the impact of bubble formation, which was not considered in the model. Higher electrolyte velocity reduces the bubble size and preserves more $H_2$ in the dissolved form[25], which serves as a reactant for the coupled hydrogenation. Indeed, our shadowgraphy experiments show decreased bubble size with increasing electrolyte velocity (Fig. S9)

due to the stronger forces (e.g., drag and shear lift) imposed on the bubbles[27]. Although our experimental velocity was limited to 0.87 cm s⁻¹ due to flow instability issues (Fig. S13a), we expect that maximum conversion will appear at a certain velocity due to the trade-off between bubble formation and hydrogen residence time. Further improvement of our coupled system, therefore, relies on more effective bubble management. Nano-structuring of the electrode, controlling the surface wettability, and adding surfactants are measures likely to further improve the $H_2$-to-MSA conversion[28–30].

Our numerical simulations also predict that the $H_2$-to-MSA conversion can be enhanced by improving the homogeneous reaction rate (Fig. 3b). The $H_2$-to-MSA conversion increases from 50% to 90% by increasing the reaction rate by a factor of 5 compared to that of the presently used Rh catalysts; 100% conversion requires a 10-fold increase of the reaction rate. Such an improvement may be achieved by designing homogeneous catalysts with a higher rate constant and/or solubility, or by enhancing the kinetics at elevated temperatures. As the reaction rate increases, hydrogenation can be completed closer to the electrode surface where the $H_2$ is generated (Fig. 3c, d), which may simplify the cell design requirements.

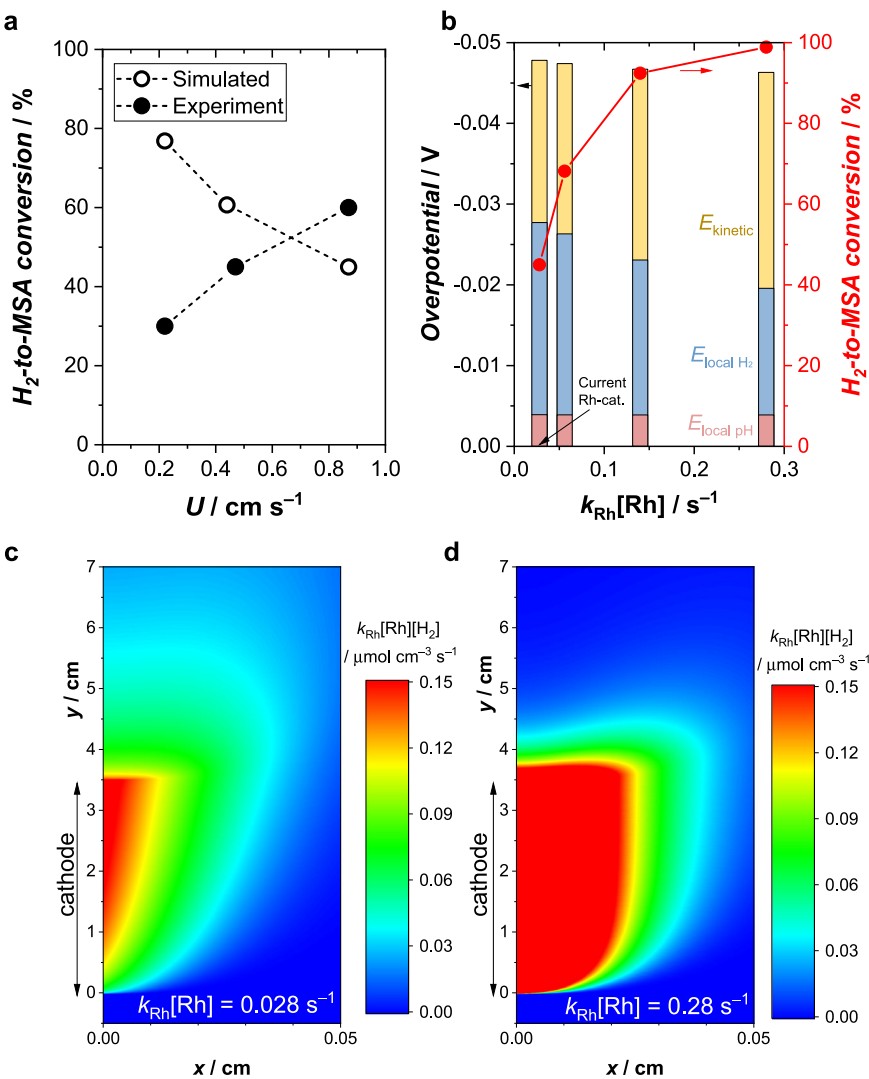

**Fig. 3 | Influence of operational and homogenous catalyst parameters. a** $H_2$-to-MSA conversion as a function of the catholyte inlet velocity ($U$) at a current density of − 2 mA cm⁻², obtained from experiments using a Pt electrode in 1 M KP_i + 0.15 M IA + 0.9 mM Rh cat. and simulations using homogeneous kinetic parameters from Fig. S1. **b** Simulated overpotential (and its breakdown into the individual contributions) and $H_2$-to-MSA conversion as a function of various homogeneous reaction rate constants at a current density of −2 mA cm⁻². The distribution of the simulated hydrogenation rate in the catholyte is shown for (**c**) $k_{Rh}[Rh]$ = 0.028 s⁻¹ and (**d**) $k_{Rh}[Rh]$ = 0.28 s⁻¹. Simulation results in **b**–**d** were performed for $U$ = 0.87 cm s⁻¹.

## Demonstration of solar-driven coupled hydrogenation

Now that the feasibility of the coupled electrochemical hydrogenation has been established, we will demonstrate solar-driven coupled hydrogenation devices using either a biased $BiVO_4$ photoanode or a GaInP/GaAs/Si photovoltaic (PV) cell as light absorbers (Figs. 4 and S10). Pt is again used as the cathode, and the irradiance spectrum of the solar simulator used in our study is shown in Fig. S11.

Because of its relatively large bandgap (2.4 eV), $BiVO_4$ is often coupled with small-bandgap semiconductors, such as silicon and III-V-based materials, in a tandem configuration[31,32]. In the $BiVO_4$/Si-based tandem concept we reported in 2019[33], two series-connected silicon heterojunction (2-SHJ) solar cells were located behind a PEC cell with $BiVO_4$ to introduce an additional voltage to drive the overall reaction. The operation voltage of this tandem device was 1.26 V (Fig. 4a), as determined from the intersection between the measured IV curve of the PEC cell and that of 2-SHJ solar cells behind $BiVO_4$[33]. In the present work, this bias voltage was applied to the $BiVO_4$ photoanode using a potentiostat to simulate the presence of the 2-SHJ solar cells. The solar-driven coupled hydrogenation performed at the expected operation voltage is shown in Fig. 4c. After an induction period of ~30 min to allow product detection by $^1$H-NMR (ca. 0.5 mmol $L^{-1}$) and/or to accumulate enough dissolved $H_2$ concentration in the solution, MSA is continuously produced from the device. The $H_2$-to-MSA conversion of this solar-driven coupled hydrogenation reaction is $53 \pm 10\%$, which is in reasonable agreement with the measurement under the dark condition shown in Fig. 2b.

Despite the relatively high $H_2$-to-MSA conversion, our demonstration using a $BiVO_4$ photoanode has several limitations. First, a continuous photocurrent decay is observed in Fig. 4c. We attribute this to the unoptimized deposition of $CoP_i$ over the 10 $cm^2$ electrode (Fig. S12). Further optimization or a switch to a more stable catalyst (e.g., $NiFeO_x$)[34] is expected to overcome this issue. In addition, as observed

in Fig. 4a, the overall device performance is limited by the modest photocurrent of the $BiVO_4$ photoanode. The latter is attributed to scale-up-related losses (e.g., ohmic losses and reactant mass transport limitations along large-area electrodes)[33,35] as well as parasitic light absorption ( < 400 nm) of our PMMA cell (Fig. S11).

To demonstrate the solar-driven coupled hydrogenation reaction without the need for an externally applied bias voltage, a GaInP/GaAs/Si tandem PV cell (4 $cm^2$) producing an open-circuit voltage (OCV) of ~2.5 V was used. Figure 4d shows the I–V curves of the PV and EC cells; the expected operating point lies at a voltage below the PV cell's maximum power point. The scheme and short-circuit unassisted operation of the device are shown in Fig. 4e, f, respectively. As expected, a steady photocurrent of ~48 mA is demonstrated, consistent with the intersection of the curves in Fig. 4d. Since the semiconductor is not in direct contact with the electrolyte, the operating current is stable, except that the OCV of the PV cell decreases slightly due to the elevated temperature of PV under prolonged solar irradiation. MSA is also successfully produced with the device (Fig. 4f), emphasizing the viability of our solar-driven coupled hydrogenation reaction. Based on the higher operation current, the amount of produced MSA is approximately 3.5 times higher than that in the demonstration with the $BiVO_4$ photoanode. However, the $H_2$-to-MSA conversion in this configuration is only $11 \pm 2\%$, significantly lower than the direct electrochemical conversion in the dark and the demonstration with the $BiVO_4$ photoanode. We attribute this low conversion to the formation of more gas bubbles at the higher operating current density. As mentioned before, we expect that higher $H_2$-to-MSA conversion can be achieved with proper bubble management. Alternatively, the homogeneous hydrogenation rate needs to be enhanced by improving its rate constant or solubility from the catalyst design or by elevating the reaction temperature to scavenge the $H_2$ produced at a higher rate. More generally, the low conversion at high current

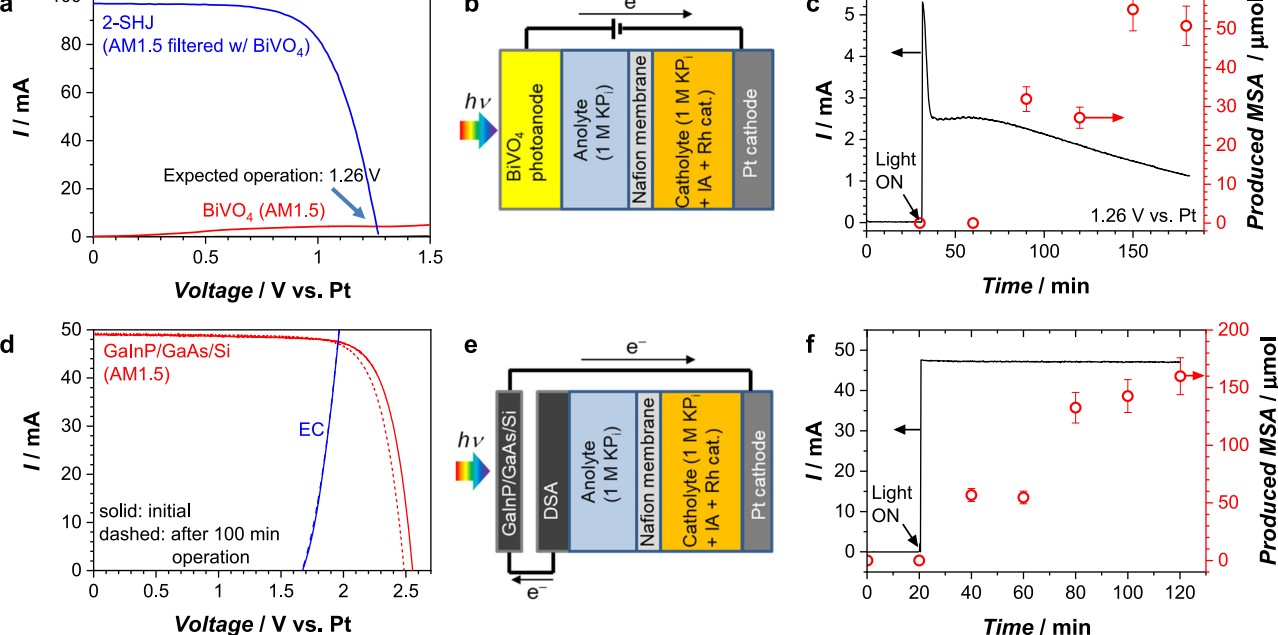

**Fig. 4 | Demonstration of solar-driven upgrading of biomass feedstock.**
**a** Current–voltage (I–V) curves of $CoP_i$/W:$BiVO_4$ (10 $cm^2$) under AM1.5 illumination and of a 2-SHJ solar cell under $BiVO_4$-filtered AM1.5 illumination. The I–V curve of the solar cell was taken from our previous report[33]. **b** Schematics, (**c**) current, and produced MSA during the operation of solar-driven coupled hydrogenation of IA using $BiVO_4$. A bias voltage of 1.26 V vs. Pt cathode was applied to simulate the 2-SHJ solar cell based on the intersection of the curves in **a**. **d** I–V curves of the GaInP/GaAs/Si tandem PV cell (4 $cm^2$) under AM1.5 illumination and of the electrochemical

cell (EC) for coupled hydrogenation of IA. Curves obtained before and after the demonstration of unassisted coupled hydrogenation for 100 min are shown in solid and dashed lines, respectively. **e** Schematics of solar-driven coupled hydrogenation of IA using PV-EC configuration, and (**f**) the total current and the amount of MSA produced during operation. Error bars in **c** and **e** were estimated as 10% based on the data points in Fig. S5a at >120 mins where the production of MSA has terminated.

densities indicates a mismatch in the rates of HER with that of hydrogenation. Indeed, $H_2$-to-MSA conversion decreases with increasing current density values (Fig. S14). At low enough current density, the $H_2$-to-MSA conversion saturates since the generated $H_2$ mainly remains as dissolved $H_2$ and the conversion is determined by the residence time and the homogenous hydrogenation rate, as discussed in Fig. 3. This suggests that photoelectrochemical devices, which typically show current densities 25-100× lower than those for electrolyzers, are particularly well-suited for coupling with homogeneous hydrogenation reactions. The 53% conversion achieved with the $BiVO_4$ photoanode is a clear illustration of this important point. Finally, although the current demonstration using the $BiVO_4$ photoanode was performed at room temperature, the electrolyte temperature will increase under prolonged solar irradiation during practical operation. This will enhance the kinetics of the homogeneous hydrogenation catalyst, resulting in improved $H_2$-to-MSA conversion as discussed in Fig. 3b. This illustrates the benefit of the efficient thermal coupling between the light absorber and the catalyst that is inherent to PEC devices (and much more difficult to achieve in wired photovoltaic-electrolyzer devices or in systems where the $H_2$ generation and hydrogenation are performed in separate reactors).

Direct detection of $H_2$ using a mass spectrometer has also been conducted using an electrochemical cell (in the dark) with a Pt cathode at operating currents that are similar to those of the coupled PEC and the PV-EC hydrogenation devices above. As shown in Table S1, the $H_2$ collection efficiencies at −3 and −50 mA are 34% and 81%, respectively. Adding these values to the respective $H_2$-to-MSA conversion efficiencies results in total Faradaic efficiency values close to 100%. In other words, there is no side reaction in the catholyte chamber other than the $H_2$ generation from the cathode and the subsequent hydrogenation of IA to MSA. The 100% total Faradaic efficiency also suggests that the presence of Rh/TPPTS does not impact the effectiveness of $H_2$ generation at the Pt cathode.

Very recently, we have performed and reported a life cycle analysis study in which it is shown that introducing the coupled hydrogenation of IA to MSA to a photoelectrochemical hydrogen production device significantly improves the net energy balance and decreases the energy payback time of the overall device[36]. In that study, we also showed that the MSA obtained from our coupled photoelectrochemical hydrogenation device requires much lower cumulative energy demand than that obtained from conventional hydrogenation using hydrogen from steam methane reforming or PV-electrolysis. To gain more insight into the benefits of our solar-driven coupled photoelectrochemical hydrogenation process, a techno-economic assessment was performed. The descriptions of our approach and parameters are provided in note S6. Only the $BiVO_4$-based PEC configuration is considered since it has been reported that solar water splitting devices or solar cells based on III-V semiconductors have higher energy demand and cost than Si-based or metal oxide-based ones[37-40].

Table 1 compares the levelized cost of hydrogen (LCOH) and the net economic profit of solar-driven coupled hydrogenation with that of solar water splitting assuming $\eta_{STH}$ of 10%. The primary cost of our photoelectrochemical device is estimated to be 367 € m$^{-2}$ (Table S3). The balance-of-system (BOS) and the operation and maintenance (O&M) components (e.g., product separation and collection, decommission) have also been taken into account in our analysis. If only solar water splitting is considered, i.e., if $H_2$ is the only product, the device is not profitable even when the lifetime is as long as 40 years (negative annual net profit in Table 1). The LCOH remains high and not competitive compared to hydrogen generated using other approaches. This is significantly improved when the PEC reaction is coupled to the hydrogenation process. Even with a $H_2$-to-MSA conversion of 11%, which is the lowest value measured in this work, the LCOH is reduced to 1.5 € kg$_{H_2}^{-1}$, which is already at the same level as that of hydrogen

**Table 1 | Levelized cost of hydrogen (LCOH) and net economic profit estimated from the techno-economic analysis (TEA) for photoelectrochemical water splitting and hydrogenation assuming $\eta_{STH}$ = 10% and the average solar insolation in Germany (3.4 kWh day$^{-1}$ m$^{-2}$)**

|  | Lifetime/ years | $H_2$-to-MSA conversion/% | LCOH/ € kg$_{H_2}^{-1}$ | Annual net profit/€ m$^{-2}$ year$^{-1}$ |
|---|---|---|---|---|
| Solar water splitting | 5 | – | 45.6 | −93 |
|  | 10 | – | 26.5 | −48 |
|  | 20 | – | 17.4 | −26 |
|  | 40 | – | 12.8 | −15 |
| Solar hydrogenation | 5 | 11 | 1.5 | 19 |
|  | 5 | 20 | −51.3 | 144 |
|  | 5 | 40 | −226.5 | 425 |
|  | 5 | 60 | −576.8 | 705 |

A device based on a tandem configuration of a $BiVO_4$ photoanode and a 2-SHJ solar cell is considered for the life cycle inventory (see Supplementary Note 6).

from steam methane reforming, even when the lifetime is only 5 years. Positive annual net profit is also achieved, indicating the profitability of the whole system. This is not surprising, because the market price of MSA is much higher than that of $H_2$. Improving the $H_2$-to-MSA conversion results in a much more favorable outcome; a device with 60% $H_2$-to-MSA conversion, which is the highest demonstrated conversion here, would generate a net profit of ~700 € m$^{-2}$ year$^{-1}$. Overall, our proposed solar-driven coupled hydrogenation concept shows significant economic advantages compared to solar water splitting.

It is noted that the PEC $H_2$ generation and the hydrogenation of IA to MSA can also be performed sequentially in separate reactors, i.e., in a non-integrated fashion. However, our integrated coupled PEC $H_2$ generation and hydrogenation system offers several advantages. Most importantly, performing the hydrogenation reaction in an integrated system will benefit from the large interfacial area between the photoelectrode surface (where the $H_2$ is generated) and the liquid electrolyte (where the hydrogenation takes place). This ensures short diffusion distances and optimal interaction between the reactants and catalysts. In non-integrated systems, significant engineering efforts would be needed to ensure homogeneous mixing and optimal interaction between the separately supplied $H_2$ and the IA + catalyst already present in the solution (e.g., by stirring vigorously, or by adding a gasification membrane); these steps can be avoided in an integrated system. One argument for separating the two processes would be that each can be individually optimized, yielding higher overall performance. However, for a homogenous reaction such as the hydrogenation of IA to MSA, all of the optimization parameters (i.e., the concentration of catalysts, feedstock/substrate, and the flow rate) can also be conveniently adjusted in the catholyte compartment of our integrated coupled PEC system. Another potential advantage of an integrated system is that the solar heating of the photoelectrode may benefit the catalytic reaction rates, although the magnitude of this effect still needs to be quantified.

We also emphasize that the hydrogenation of IA to MSA, despite being energetically attractive, is only used here as a model reaction. The coupled PEC + hydrogenation concept offers great flexibility towards a variety of hydrogenation reactions (e.g., levulinic acid to -γ-valerolactonate[41], acetone to isopropanol[42]) since the hydrogenation reaction itself can be done by homogeneous catalysis and does not need to be performed (photo)electrochemically. This also enables enantioselectivity towards specific chiral products, by introducing the appropriate homogeneous catalysts. It is even possible to rapidly change the hydrogenation reactions on demand by simply replacing the solutions that contain the reactants and homogeneous catalysts,

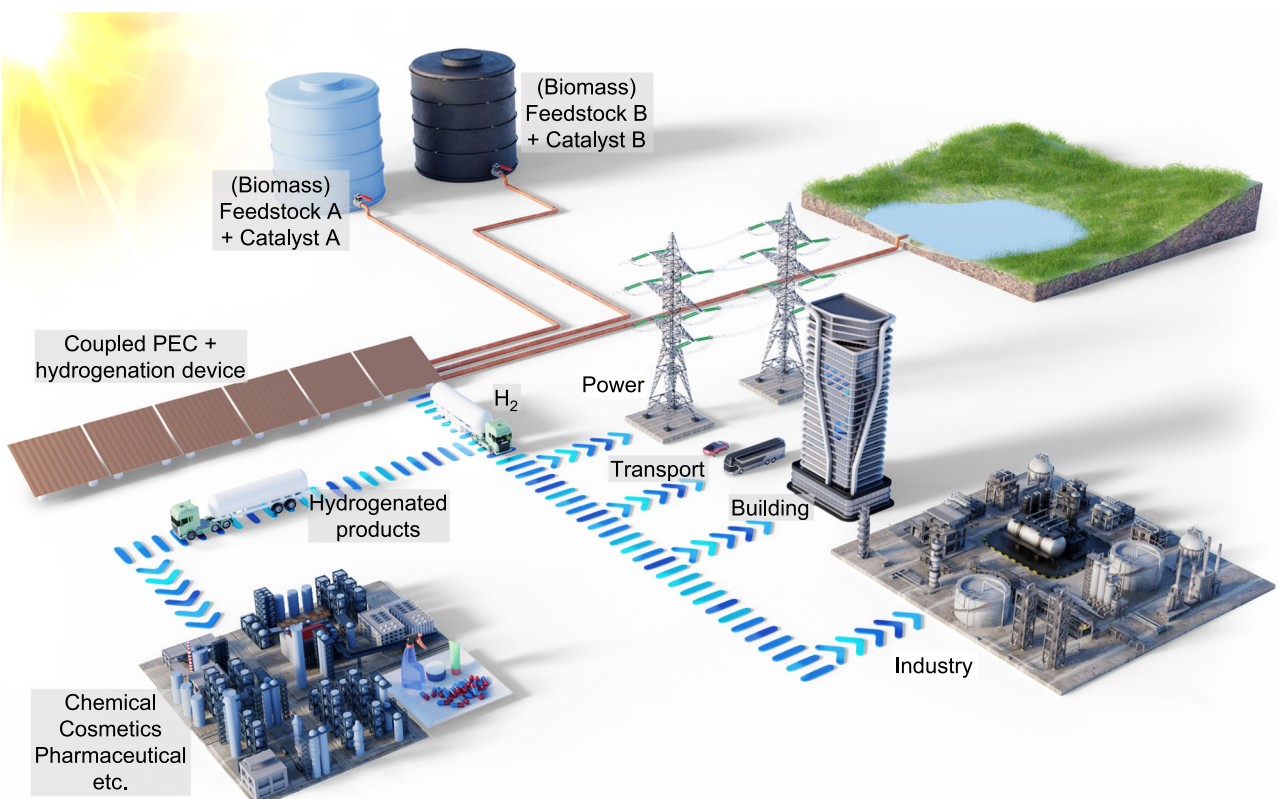

**Fig. 5 | Illustration of the solar-driven coupled photoelectrochemical and hydrogenation plant.** The coupled photoelectrochemical and hydrogenation device uses sunlight to generate hydrogen and partially consumes the generated hydrogen in situ to hydrogenate (biomass) feedstock using a homogeneous catalyst. The hydrogen can then be utilized for energy storage and regeneration for power, transport, building, and other industrial applications. The hydrogenated products, which are valuable chemicals, can be utilized in various processes e.g., chemical, cosmetics, and pharmaceuticals production. The coupled concept allows flexible control over increasing or decreasing the conversion efficiency towards the hydrogenated products by simply adjusting the supply of feedstock and homogeneous catalyst concentrations. This allows the plant to adjust the production yield in response to changes in the product demand. In addition, different hydrogenation products can be generated by simply switching the supply of the feedstock and the catalyst (e.g., as illustrated between "Feedstock A + Catalyst A" to "Feedstock B + Catalyst B") without the need to change the photoelectrochemical device.

without having to change the (photo)electrochemical device itself (Fig. 5). Another consideration is the thermodynamics of the hydrogenation reactions. The redox potential of our present reaction (i.e., IA to MSA) is +0.42 V vs. RHE, which causes spontaneous hydrogenation using the in situ generated $H_2$ with not insignificant energy losses. Other reactions with redox potentials closer to that of HER can be considered to minimize the energy loss, although it may also reduce the driving force for the homogeneous reaction and lead to slower reaction rates. Despite having used relatively large photoelectrodes in this study (10 cm² vs. ≤ 1 cm² in most lab-scale studies), further scale-up efforts to a larger area and/or higher production yield are needed for the practical implementation of the coupled concept. These efforts will benefit significantly from the recent progress on scaling up solar-driven water splitting devices[33,35,43,44], since the faradaic reactions in our coupled system are essentially the same as those in solar water splitting. Finally, the separation of products and homogeneous catalysts is not yet experimentally considered in our study. Recent techniques, such as micellar-enhanced ultrafiltration and cloud point extraction[21,45], are likely to be applicable to our coupled reaction concept and need to be explored in future studies.

In summary, we have demonstrated the concept of solar-driven upgrading of biomass-derived feedstocks by in situ utilization of (photo)electrochemically generated $H_2$ to perform a hydrogenation reaction over homogeneous catalysts dispersed in the catholyte. By taking the hydrogenation of IA to MSA as a model reaction and Rh-TPPTS as the homogenous catalyst, we have shown that the coupled process is more stable and far less sensitive to deactivation than the direct heterogeneous electrochemical hydrogenation. Through a combination of multiphysics simulations and experiments, we found that the catalytic reaction rate and bubble management are critical parameters in controlling hydrogenation and product conversion. A $H_2$-to-MSA conversion of up to 60% was achieved with coupled electrochemical hydrogenation (dark), and solar-driven coupled hydrogenation devices based on PEC and PV-EC approaches were successfully demonstrated with a $H_2$-to-MSA conversion of up to 53%. Using a photoanode leads to higher conversion than using an electrolyzer since the reaction rate of the former matches better with the hydrogenation rate. A techno-economic analysis shows that by coupling the PEC water splitting reaction with the in situ catalytic hydrogenation of IA, the levelized cost of hydrogen could be dramatically reduced to the point that it becomes competitive even with generating hydrogen from steam methane reforming. Combined with the favorable life-cycle net energy balance we recently reported[36], these results clearly demonstrate the overall benefit of coupling a hydrogenation reaction with a PEC $H_2$ production. Finally, the coupled PEC/catalysis approach can be used for a wide variety of chemical transformations through the appropriate design of homogeneous catalysts and offers a promising pathway to make solar hydrogen production economically feasible.

## Methods
### Homogeneous hydrogenation reactions
Semi-batch homogeneous hydrogenation reactions of itaconic acid (VWR, ≥99% purity) to methyl succinic acid were performed in a double-walled glass reactor equipped with a gas-dispersion stirrer. The

reactions were carried out at $T = 25\,°C$, $p = 0.11\,MPa$, and $n$ (stirrer speed) = 1550 rpm. During the reactions, the pressure was kept constant using a pressure controller (Bronkhorst). The volume of the consumed hydrogen was measured with a flow meter (Bronkhorst). For a typical reaction, 100 mL of an aqueous solution containing the required amount of itaconic acid ($c_{IA} = 2.5 - 20\,g\,L^{-1}$) was placed into the reactor, and the reactor was sealed. The thermostat (Haake, F6C25) was set to 25 °C, and the reactor was evacuated and purged under stirring with nitrogen ($n = 1550\,min^{-1}$) three times. Then, the freshly prepared homogeneous hydrogenation catalyst Rh/TPPTS ($c_{Rh} = 100 - 900\,\mu M$) was added through a septum, and the reactor was again evacuated and purged with nitrogen three times. The stirrer speed was lowered to 800 $min^{-1}$, and the temperature was equilibrated for 20 min. The stirrer speed was raised to 1550 $min^{-1}$, the reactor was evacuated, and then the stirring was terminated. The reactor was filled with hydrogen until a total pressure of 0.11 MPa was reached. The reaction was initiated by starting the stirrer, and the time and the consumed hydrogen were recorded by an Excel macro. From the consumed hydrogen $V(t)$, the conversion $X_{IA}$ of itaconic acid was calculated using Eq. 1.

$$X_{IA}(t) = \frac{V(t)}{V_{total}} \qquad (1)$$

where $V_{total}$ is the total amount of consumed hydrogen. From the conversion profile, the initial reaction rate $r_0$ for $X_{IA}$ equal to 10% was calculated from Eq. 2

$$r_0 = c_{IA,0} \cdot \left(\frac{dX}{dt}\right)_{X=10\%} \qquad (2)$$

where $c_{IA,0}$ is the initial concentration of itaconic acid and $dX/dt$ is the initial slope of the $X$-$t$-curve.

The Rh/TPPTS catalyst complex was prepared from [Rh(cod)Cl]$_2$ (TCI) as the Rh precursor and TPPTS as the phosphine ligand using a rhodium to phosphine (Rh/P) ratio of 1/7. The concentration of TPPTS is 24 wt% in water. The Rh precursor was placed in a Schlenk tube together with the TPPTS solution, and the gas in the headspace was replaced by argon. The solution was stirred overnight to form the catalyst complex, indicated by a color change from yellow to red.

### Preparation of the electrodes

**Pt cathode.** In all, 200 nm of Pt was evaporated on TEC 7™ FTO substrates ($5 \times 5\,cm^2$) with 5 nm Ti as an adhesion layer. Depositions were done by electron beam evaporation (Telemark) in a customized high vacuum deposition chamber pumped by a typical dry turbo molecular pumping set with a typical base pressure of $2 \times 10^{-7}$ mbar. Deposition rates of 0.15 nm $s^{-1}$ for Ti (0.4 kW e-beam power) and 0.65 nm $s^{-1}$ for Pt (2 kW) were used and controlled during deposition using a quartz crystal microbalance. The Pt electrode was electrochemically cleaned in 0.5 M KP$_i$ by cyclic voltammetry between $-0.03$ and 1.72 V vs. RHE at a scan rate of 100 mV $s^{-1}$ for 30–60 cycles.

### BiVO$_4$ photoanode

To reduce ohmic losses due to the FTO substrate ($5 \times 5\,cm^2$), Ni lines were electrochemically deposited on the substrate as described in our previous report[33]. In short, the cleaned FTO substrates were masked with Kapton® tape followed by chemical reduction process immersing it in an aqueous solution containing 1.0 M glycine (≥99%, Aldrich), 0.5 M FeSO$_4$·7H$_2$O (≥99%, Aldrich) at pH 2.5 for 3 min. Zn powders (mesh 100) were uniformly dispersed on FTO (FTO-side facing upward) for 10 min. The electrochemical deposition of Ni lines on the chemically reduced FTO were performed by a chronopotentiometry at $-5$ mA $cm^{-2}$ with stirring for 15 min. The deposition bath was composed of 1.14 M NiSO$_4$·7H$_2$O (≥98%, Aldrich), 0.16 M NiCl$_2$·6H$_2$O (≥98%,

Aldrich), and 0.73 M H$_3$BO$_3$ (≥99.5%, Aldrich) at 50 °C. After the deposition, the Kapton® tape was removed, and FTO with Ni lines were ultrasonically cleaned in deionized water, acetone, and ethanol, each for 60 s. Uniform deposition of 1% W-doped BiVO$_4$ and a SnO$_2$ hole blocking layer was obtained by spray pyrolysis as described in our previous report[33]. First, a - 10 nm thick SnO$_2$ layer was deposited by spray pyrolysis using a solution of 0.1 M SnCl$_4$ in ethyl-acetate. Subsequently, -200 nm thick BiVO$_4$ with 1% W doping was deposited by spray pyrolysis using a precursor solution 4.44 mM Bi(NO$_3$)$_3$·5H$_2$O (98%, Sigma-Aldrich), 4.396 mM VO(C$_2$H$_7$O$_2$)$_2$ (99%, Alfa Aesar), and 0.044 mM W(C$_2$H$_5$O)$_6$ from 5% w/v in ethanol (99.8%, Alfa Aesar). The solvent consists of a 10 vol% of acetic acid in absolute ethanol. The substrate was maintained at 450 °C during the deposition. The CoP$_i$ OER catalyst was deposited photoelectrochemically in a commercial Micro Flow Cell (ElectroCell) at the flow rate of 60 mL $min^{-1}$ under AM1.5 G illumination at a constant applied potential of 1 V vs. RHE in 1 M KP$_i$ solution containing 1 mM Co(NO$_3$)$_2$·6H$_2$O (>99%, EMSURE®), which is named CoP$_i$/W:BiVO$_4$[33]. The total charge passed during the deposition was 0.4 C $cm^{-2}$. A conditioning step was done after the deposition by performing cyclic voltammetry between 0.2–1.7 V vs. RHE with a scan rate of 10 mV $s^{-1}$ for 5 cycles in the dark and 5 cycles under illumination[33].

### (Photo)Electrochemical measurements

All the (photo)electrochemical measurements were performed using a commercial Micro Flow Cell (ElectroCell) with a cation exchange membrane (NRE-212, Nafion™, thickness 0.002 inches) located between the anolyte and catholyte chambers (see Fig. S2). The geometric active area of the electrodes was 10 cm$^2$ after the electrodes were covered with gaskets. The anolyte and catholyte solutions (60 mL) were continuously degassed by Ar gas before and during measurements in order to prevent O$_2$ contamination. These solutions were continuously circulated by peristaltic pumps (TBE/200, MDX Biotechnik International GmbH) with a flow rate range of 15–60 mL $min^{-1}$, which corresponds to a velocity range of 0.2–0.9 cm $s^{-1}$ in the electrochemical cell. In this velocity range, the velocity seemed to be stable, and the mass-transport limiting current of Fe(CN)$_6^{3-}$ reasonably agreed with the simulated one, which indicates that the flow is laminar and stable (Fig. S13). In all, 1 mL of liquid samples were collected through a septum and the products were quantified by $^1$H-NMR. The H$_2$-to-MSA conversion, $\eta_{H2\text{-to-MSA}}$, was estimated based on $^1$H-NMR product quantification using Eq. 3, assuming that the Faradaic efficiency for H$_2$ is 100% on the Pt cathode.

$$\eta_{H2-to-MSA} = \frac{r_{MSA}}{j_{app}A/2F} \qquad (3)$$

where $r_{MSA}$, $j_{app}$, and $A$ are the production rate of MSA, the applied current density, and the electrode area, respectively. Note that, for the coupled approach, we simply report the H$_2$-to-MSA conversion instead of the Faradaic efficiency as a measure of product selectivity. This is because in contrast to direct heterogeneous electrochemical reactions, the predominant Faradaic reaction on Pt in our proposed system is still the production of H$_2$.

1 M potassium phosphate (KP$_i$) buffer solutions (pH = 7) were prepared from KH$_2$PO$_4$ (≥ 99.0 %, Sigma-Aldrich) and K$_2$HPO$_4$·3H$_2$O (≥99.0 %, Sigma-Aldrich) and used as the anolyte and the catholyte. Because adding 0.15 M of itaconic acid (≥99%, Aldrich) shifts the pH of the solutions to more acidic values, KOH (85-100.5%, Merck) was added to maintain the pH around 7. The homogeneous catalyst solution was prepared from Rh(COD)Cl$_2$ (98%, Aldrich) and 3,3',3''-phosphanetriyltris(benzenesulfonic acid) trisodium salt (TPPTS, Sigma-Aldrich) dissolved in water ($c_{TPPTS} = 24$ wt%) and stabilized overnight under Ar flow. Prior to the measurements, the homogeneous catalyst solution was mixed with KP$_i$ solution containing IA. All the precursors

were weighed to obtain the desired final composition, i.e., 1 M $KP_i$, 0.15 M IA, 0.3 M KOH, 0.89 mM $Rh(COD)Cl_2$, and 6.7 mM TPPTS. The water used in all experiments was obtained from a Milli-Q Integral system with a resistivity of 18.2 MΩ cm.

All the electrochemical measurements were performed using a VersaSTAT 3 potentiostat/galvanostat (AMETEK). The uncompensated resistance ($R_u$) was obtained from impedance measurements, and $iR_u$ corrections were performed to the applied voltage unless otherwise stated. Electrochemical reactions on Pt were studied in a three-electrode configuration with Ag/AgCl (3.4 M KCl) reference electrode (LF-1, Innovative Instruments Ltd.) located in the catholyte chamber. Because of the oxidation of organic substance on Pt, the potential sweep is limited up to 0.8 V vs. RHE in the presence of organic additives. A dimensionally stable anode (DSA®) was used as an anode unless otherwise specified. Solar-driven demonstrations were performed with a two-electrode configuration with a solar simulator (class AAA, WXS-100S-L2H AM 1.5GMM, WACOM) whose spectrum is shown in Fig. S11. Control experiments confirmed that the production of MSA shown in Fig. 4 is not caused by a direct photochemical reaction of the Rh catalysts (see note S5). $R_u$ in 1 M $KP_i$ was approximately 0.5 and 1.6 Ω in three- and two-electrode configurations, respectively. A triple-junction GaInP/GaAs/Si PV cell (4 cm$^2$), as previously reported in the literature[46], was obtained from Fraunhofer ISE.

Shadowgraphy measurements to observe $H_2$ bubbles were performed using the same cell and configuration as mentioned above, but a thin Pt (~5 nm) was instead deposited on an FTO substrate since adequate sample transparency was needed. Bubble image capture was done with the same WACOM solar simulator as the light source and a camera system from LaVision® (2752 × 2200 pixels, frame rate 25 Hz). The captured images were processed, and the $H_2$ bubbles were quantified using a bubble shadowgraphy module in the DaVis 10 software provided by LaVision®. To minimize the contribution of nucleating bubbles and bubbles agglomerated at the electrode's surface to the overall statistics, bubbles with diameters smaller than 100 μm and larger than 300 μm were not considered in the analysis.

### Product quantification

The amounts of itaconic acid (IA) and methyl succinic acid (MSA) after (photo)electrochemical hydrogenation reactions were determined from $^1$H-NMR measurements (Bruker Avance, 400 MHz). Liquid samples were taken from the catholyte periodically and diluted with $D_2O$. The spectra of the pure solutions (IA, MSA) and mixture solutions are shown in Fig. S15. The most intense $^1$H-NMR peaks for IA and MSA were selected for the calculations. An error of ~10% was estimated from the measurements after the direct hydrogenation on Pt completely ceased (see Fig. S5a). $H_2$ detection measurements were performed by inserting a micro-capillary tube connected to a mass spectrometer (HPR-40, HIDEN Analytical) into the catholyte reservoir. Calibration was performed using a gaseous flow of pre-mixed 2.0 % $H_2$ in argon. UV-vis transmission spectroscopy was performed using a white light source (deuterium-halogen lamp, DH-2000-BAL, Ocean Optics) and a CCD spectrometer (Maya 2000-Pro, Ocean Optics) coupled with optical fibers and collimators.

### Numerical simulations

Details of the numerical simulation are described in Supplementary Note 4. In short, the 2D geometry of the catholyte chamber was defined based on the actual geometry of the electrochemical flow cell used as shown in Fig. S16. A single-phase fully developed laminar flow was assumed within the channel. Concentration-dependent Butler-Volmer equation was solved on the cathode with the consumption of $H^+$ and the production of $H_2$. The membrane provides $H^+$ to counterbalance the consumption of $H^+$ on the cathode. The transport of

chemical species was simulated by the Nernst-Planck equation with charge neutrality and buffer equilibrium. In the electrolyte domain, a homogeneous reaction is also introduced as shown below.

$$r = k_{Rh}c_{Rh}c_{H_2} \tag{4}$$

$r$, $k_{Rh}$, and $c_i$ are the homogeneous reaction rate, the rate constant of the Rh catalyst, and the concentration of species, i, respectively. $r$ determines the consumption of IA and $H_2$, and the production of MSA in the electrolyte. The $H_2$-to-MSA conversion was defined as the molar flux of MSA at the outlet divided by the theoretical production rate at a given current density. All the parameters used are shown in Table S2. Steady-state simulations were performed with COMSOL Multiphysics® using PARDISO general solver. Relative tolerance of 0.001 was applied as the convergence criterion.

### Reporting summary

Further information on research design is available in the Nature Portfolio Reporting Summary linked to this article.

### Data availability

All data supporting the findings of this study are available within the main text and the Supplementary Information. Source data of the figures in the main text and Supplementary Information are provided with this paper. Source data are provided with this paper.

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

## Acknowledgements

This research was supported by the Deutsche Forschungsgemeinschaft (DFG, German Research Foundation) under Germany's Excellence Strategy – EXC 2008/1 (UniSysCat) – 390540038 and from the German Helmholtz Association–Excellence Network–ExNet-0024-Phase2-3. This work was carried out with the support of the Helmholtz Energy Materials Foundry (HEMF), a large-scale distributed research infrastructure funded by the German Helmholtz Association. B.R. and F.F.A. acknowledge support from the European Union's Horizon 2020 Research and Innovation Action program under the Grant Agreement No. 862192 (SunCoChem). We acknowledge Dr. Markus Feifel and Dr. Frank Dimroth from the Fraunhofer ISE for providing the GaInP/GaAs/Si tandem PV cell used in this study, Dr. Peter Bogdanoff for the assistance in the $H_2$ detection measurements, Karsten Harbauer for the preparation of the Pt/

Ti/FTO electrodes, as well as Zishuo Li for his initial assistance with the techno-economic analysis.

## Author contributions
R.v.d.K., R.S. and F.F.A. conceived the idea, acquired funding, and supervised the project. K.O. performed the electrochemical measurements and the numerical simulations. M.S. and T.A.T. performed the homogeneous hydrogenation reactions, NMR measurements, and their analysis. X.Z. and F.F.A. performed the techno-economic study. B.R. and I.Y.A. prepared the photoelectrodes. K.O. and B.R. performed the demonstration of the solar-driven coupled hydrogenation devices. B.R. performed the H2 detection and shadowgraphy experiments. K.O., M.S. and F.F.A. wrote the manuscript. All authors contributed to the scientific discussion and the final version of the manuscript.

## Funding

## Competing interests
The authors declare no competing interests.
