## [Peer Review File · Nature Communications]

Solar-driven upgrading of biomass by coupled hydrogenation using in situ (photo)electrochemically generated H₂REVIEWER COMMENTS

Reviewer #1 (Remarks to the Author):

This manuscript coupled the photoelectrochemical production of hydrogen with the Ru/TPPTS-catalyzed hydrogenation of itaconic acid (IA) to methyl succinic acid (MSA), offering an alternative and green pathway to fossil fuel-based chemical production processes. The achievement "continuous hydrogenation with up to 60% H₂-to-MSA conversion" is attractive, and this concept is of great potential. However, as bifunctional catalysts or tandem catalysts are extensively developed and widely applied in various fields, I am not sure that "coupling production of hydrogen with a hydrogenation reaction" was not reported in the past. Besides, some control tests should be added to reveal the effects of Ru/TPPTS on the production of hydrogen. In my opinion, this manuscript can be published after major revisions. The comments are listed as follows.

1. The authors stated that this manuscript is "the first time an integrated solar-driven device in which the photoelectrochemical production of hydrogen is coupled to a catalytic hydrogenation reaction". However, this strategy is quite similar to bifunctional catalysts or tandem catalysts. I am not sure if the so-called "first time" in this manuscript is accurate.
2. Due to "the blockage of the active sites by Rh/TPPTS catalysts and excess of TPPTS ligands", Ru/TPPTS can prevent the direct electrochemical IA hydrogenation on Pt. What is the effect of Ru/TPPTS on the photoelectrochemical production of hydrogen? How serious is this effect?
3. Although this manuscript stated that "the Rh catalyst prevents deactivation", which led to stable hydrogenation (Fig. 1b). However, 90 min is too short to show the stability. How about the stability in a longer period? In addition, the rate of hydrogenation seems speeding up after 60 min. Why? Will the rate continue speeding up while extending the reaction period? As a control, the stability of Ru/TPPTS-catalyzed IA hydrogenation assisted by other H₂ sources should be provided.
4. As indicated, the rate of H₂-to-MSA conversion depends on the rate matching between HER and hydrogenation. Therefore, the value of applied bias should largely affect the conversion rate, because HER largely depends on bias while hydrogenation not (in principle). Could you provide this data? This will help reader to better understand the system.
5. Will the applied bias influence the hydrogenation reaction? If yes, please provide data to reveal the influence. If not, why not separate the HER and hydrogenation reaction? These two reactions can be carried out in two reactors. The H₂ gas can be guided to another reactor for hydrogenation reaction. Then, we can use more optimized reaction conditions for both HER and hydrogenation to improve the efficiency.
6. As the work of Holzhäuser et al. in 2017 was mentioned, why the authors used Pt rather than Pb? I wonder if the direct and Ru/TPPTS-catalyzed IA hydrogenation can be coupled on Pb system.
7. In Fig. 4, it is difficult to recycle the homogeneous catalysts in Feedstock A and B. The separation of catalysts and products from the obtained mixture may be costly and challenging.

Reviewer #2 (Remarks to the Author):

This article reports on the in situ use of (photo)electrochemically generated H₂ for the homogeneous hydrogenation of itaconic acid to methyl succinic acid. ~60% of the produced hydrogen can be utilized under optimized condition. Additionally, a techno-economic assessment of the proposed device further indicated the benefit of coupling solar hydrogen production to a chemical transformation. This makes full use of the cathode in the solar electrochemical cell. The idea is novel and the article is well organized. It is suitable for publication in NC. The following point might be considered before publication.

- (1) More description about the catalytic mechanism of Rh/TPPTS catalyst in the system is needed.
- (2) Direct electrochemical IA hydrogenation to MSA demonstrated by Holzhäuser has low stability.

How about the direct electrochemical IA hydrogenation with Rh/TPPTS catalyst at low current density (such as 2 mA/cm²)?

(3) It is predicted that the use Pt as a model cathode in this study can be replaced with alternative earth-abundant HER catalysts such as NiMo, transition metal dichalcogenides, and phosphides. It would be great to provide some preliminary results about it?

(4) Does the potential of cathode affect the utilization of in-situ produced hydrogen?

(5) How about the selectivity in the reaction of IA hydrogenation to MSA for this system?

(6) In the techno-economic assessment, it would be great to include the technology of PV-EC combined with traditional hydrogenation to highlight the (photo)electrochemically generated H₂ for the homogeneous hydrogenation.

Reviewer #3 (Remarks to the Author):

The manuscript of Abdi and coworkers entitled "Solar-drive upgrading of biomass by coupled hydrogenation using in situ (photo)electrochemically generated H₂" proposes the use of the H₂ generated either electrochemically or photoelectrochemically to turn itaconic acid into methyl succinic acid using a homogeneous catalyst dissolved in the electrolyte. The manuscript is interesting, in the sense that it opens a new gateway to utilize PEC devices in a more economically-competitive fashion. Indeed, it is a follow up paper from a previous article these authors published in this same journal (DOI: 10.1038/s41467-023-36574-1), cited as REF. 32. While in that previous report the authors mainly proposed the concept of the "coupled hydrogenation" and the economic viability of this "indirect" hydrogenation approach, here, the authors describe the experimental evidence and elaborate on the aspects that control the performance. Overall, the results and conclusions align well with the requirements for publication in nature communications, and therefore I would support the acceptance after completing a minor revision to clarify some aspects:

(1) In page 7 (Line141), the authors claim that "cathodic shoulder" is indicative of the direct heterogeneous hydrogenation. What are the authors referring to? The authors should provide some experimental evidence that supports their claim on the "deactivation" of the heterogeneous reaction. If not, it seems quite hasty to rule out the participation of such a reaction.

(2) The authors conclude that the rate of H₂ production critically affects the H₂ concentration in the electrolyte and hence the conversion rate of the itaconic acid. While the authors explored the effect of the catholyte flow and the H₂ bubble size, did the authors evaluate the effect of the current density on the selectivity? It would be interesting to show how the selectivity changes as a function of the current density.

(3) The LCOH analysis provided by the authors did not consider the separation process of the product. While they propose some techniques this was not explicitly contemplated in the LCOH. Could the authors estimate the cost for the separation process to provide a more complete picture of the whole process?

Response to reviewers' comments

Manuscript ID: NCOMMS-23-13885-T

Title: Solar-driven upgrading of biomass by coupled hydrogenation using in situ (photo)electrochemically generated H₂

In this response letter, the reviewers' comments to our original manuscript are provided in **black**, and our point-by-point responses as well as the corresponding changes to the manuscript are shown in **blue**.

Reviewer #1 (Remarks to the Author):

This manuscript coupled the photoelectrochemical production of hydrogen with the Rh/TPPTS-catalyzed hydrogenation of itaconic acid (IA) to methyl succinic acid (MSA), offering an alternative and green pathway to fossil fuel-based chemical production processes. The achievement “continuous hydrogenation with up to 60% H₂-to-MSA conversion” is attractive, and this concept is of great potential. However, as bifunctional catalysts or tandem catalysts are extensively developed and widely applied in various fields, I am not sure that “coupling production of hydrogen with a hydrogenation reaction” was not reported in the past. Besides, some control tests should be added to reveal the effects of Rh/TPPTS on the production of hydrogen. In my opinion, this manuscript can be published after major revisions. The comments are listed as follows.

1. The authors stated that this manuscript is “the first time an integrated solar-driven device in which the photoelectrochemical production of hydrogen is coupled to a catalytic hydrogenation reaction”. However, this strategy is quite similar to bifunctional catalysts or tandem catalysts. I am not sure if the so-called "first time" in this manuscript is accurate.

Response: We appreciate the comment of the reviewer regarding our claim. While the concept of our device shares some similarities with bifunctional or tandem catalysis, to the best of our knowledge, we are not aware of any reports of coupled photoelectrochemical production of hydrogen with homogeneous hydrogenation. We, therefore, believe our initial statement remains true. Nevertheless, in order to be more cautious with our claim, we have modified the statement by removing the “the first time” phrase from our sentence.

Associated changes to the manuscript:

- Page 3: “In this study, we demonstrate ~~for the first time~~ an integrated solar-driven device in which the photoelectrochemical production of hydrogen is coupled to a catalytic hydrogenation reaction.”

2. Due to “the blockage of the active sites by Rh/TPPTS catalysts and excess of TPPTS ligands”, Rh/TPPTS can prevent the direct electrochemical IA hydrogenation on Pt. What is the effect of Rh/TPPTS on the photoelectrochemical production of hydrogen? How serious is this effect?

Response: We first re-emphasize that the Pt electrode was also used as the cathode in our coupled photoelectrochemical device. The impact of Rh/TPPTS on direct electrochemical hydrogenation is therefore not different between our coupled electrochemical and coupled photoelectrochemical device. Regarding the impact on the production of hydrogen, as reported on pages 14-15 and Table S1, we have

confirmed that the total Faradaic efficiency of our coupled device is very close to 100%. This means that there is no side reaction, and the presence of Rh/TPPTS does not impact the H₂ generation at the Pt cathode.

Associated changes to the manuscript:

- Page 15: “In other words, there is no side reaction in the catholyte chamber other than the H₂ generation from the cathode and the subsequent hydrogenation of IA to MSA. The 100% total Faradaic efficiency also suggests that the presence of Rh/TPPTS does not impact the effectiveness of H₂ generation at the Pt cathode.”

3. Although this manuscript stated that “the Rh catalyst prevents deactivation”, which led to stable hydrogenation (Fig. 1b). However, 90 min is too short to show the stability. How about the stability in a longer period? In addition, the rate of hydrogenation seems speeding up after 60 min. Why? Will the rate continue speeding up while extending the reaction period? As a control, the stability of Rh/TPPTS-catalyzed IA hydrogenation assisted by other H₂ sources should be provided.

Response: We thank the comment from the reviewer regarding the stability of the hydrogenation reaction. We have indeed measured the production of MSA for a longer period and shown the comparison in Figure S5, which is reproduced in this response letter as Figure R1. We note that the flow rate was varied for comparison in the data shown in panel (b), and the data shown in Fig. 1b was taken from the results between 180 and 270 min. Although the rate of hydrogenation seems to speed up at the end, it is still within the error range. As shown in Fig. R1, the reaction rate is relatively constant at any given operating condition.

Figure R1. The amount of MSA produced (a) without and (b) with the Rh/TPPTS catalyst at -2 mA cm^{-2} . The velocity of the electrolyte is (a) 0.87 cm/s and (b) $0.22 - 0.87 \text{ cm/s}$.

Regarding the stability of the Rh/TPPTS catalyst, the amount of produced MSA was limited to $800 \mu\text{mol}$ (see Fig. R1) in the case of our coupled hydrogenation device due to the limited generation of H₂ from the Pt cathode. However, further conversion using Rh/TPPTS catalysts was, indeed, demonstrated in experiments using other H₂ sources. This is shown in Fig. S1, which is reproduced in Fig. R2 in this response letter. Continuous hydrogenation using an external H₂ feed was obtained until the reactant (IA) in the solution was fully depleted. Considering the solution volume of 100 mL and IA concentration of 78 mM (Fig. R2a), the total produced MSA reached 7.8 mmol (i.e., one order of magnitude higher than our coupled hydrogenation device) without having any noticeable deactivation. Further discussion on the long-term operation and recyclability tests for the hydrogenation of IA or dimethyl itaconate using Rh/TPPTS catalyst by feeding external H₂ have been reported in our previous reports (J. S. Milano-Brusco, et al., *Ind. Eng. Chem. Res.*, 2008, 47, 7586; M. Schmidt, et al., *Ind. Eng. Chem. Res.*, 2019, 58, 2445).

Figure R2. (a) Cumulative hydrogen consumption and the corresponding conversion of IA with various concentrations of Rh/TPPTS catalyst, c_{cat} , ($n_{\text{Rh}}:n_{\text{TPPTS}} = 1:7$), and $c_{\text{IA}} = 78 \text{ mM}$ during the homogeneous hydrogenation in a semi-batch reactor. The initial reaction rates of the hydrogenation of itaconic acid are plotted with the variation of (b) Rh/TPPTS concentration (c_{IA} was kept constant at 78 mM), and (c) IA concentration (c_{cat} was kept constant at 0.9 mM). (d) The conversion of IA in pure H_2O and KP_1 solutions ($c_{\text{cat}} = 0.9 \text{ mM}$, and $c_{\text{IA}} = 78 \text{ mM}$). In all experiments, the temperature ($T = 25^\circ\text{C}$), total pressure (0.11 MPa), and stirrer speed (1550 rpm) were kept constant.

Associated changes to the manuscript:

- Page 8: “Although the demonstrated total production of MSA using our coupled electrochemical hydrogenation was limited to 800 μmol in the present test (Fig. S5b), further production should be possible without any noticeable deactivation until the IA concentration fully depletes in the solution. Indeed, our experiments in a semi-batch reactor using an external feed of H_2 (Fig. S1) show no deactivation of hydrogenation rate up until the IA is fully depleted. The total produced MSA in this case was estimated to be 7.8 mmol. Further recyclability tests of the Rh/TPPTS catalysts have been discussed in our previous reports.^{9,18}”

4. As indicated, the rate of H_2 -to-MSA conversion depends on the rate matching between HER and hydrogenation. Therefore, the value of applied bias should largely affect the conversion rate, because HER largely depends on bias while hydrogenation not (in principle). Could you provide this data? This will help reader to better understand the system.

Response: Since the H_2 -to-MSA conversion is largely dependent on the matching rate between HER and hydrogenation, the important parameter that affects the conversion is the HER current density. The applied bias of course affects the current density, but its impact on the overall H_2 -to-MSA conversion is indirect.

We agree with the reviewer that providing the relationship between the HER rate (current density) and the H₂-to-MSA conversion will help the readers to better understand the system. We have indeed reported the H₂-to-MSA conversion of our system at 0.3 and 5 mA/cm² in Table S1. We have now combined the result from the measurement at 2 mA/cm² and plotted the H₂-to-MSA conversion as a function of the current density in Fig. R3. As expected, the H₂-to-MSA conversion decreases with increasing current density because of the increasing mismatch between the HER rate and the hydrogenation rate.

Figure R3. H₂-to-MSA conversion values obtained from a Pt electrode (in dark conditions) as a function of the current density. The electrolyte was 1 M KP_i with 0.15 M itaconic acid (IA) and 0.9 mM Rh/TPPTS catalyst.

Associated changes to the manuscript:

- Figure R3 is added to the Supporting Information as Figure S13.
- Page 15: “More generally, the low conversion at high current densities indicates a mismatch in the rates of HER with that of the hydrogenation. **Indeed, H₂-to-MSA conversion decreases with increasing current density values (Fig. S13).**”

5. Will the applied bias influence the hydrogenation reaction? If yes, please provide data to reveal the influence. If not, why not separate the HER and hydrogenation reaction? These two reactions can be carried out in two reactors. The H₂ gas can be guided to another reactor for hydrogenation reaction. Then, we can use more optimized reaction conditions for both HER and hydrogenation to improve the efficiency.

Response: As mentioned in our response to comment #4 above, the applied bias will not directly influence the hydrogenation reaction as the hydrogenation reaction occurs homogeneously in the catholyte solution. The applied bias will, however, affect the current density, which will in turn influence the rate of hydrogenation. One important benefit of integrating the photoelectrochemical HER and the hydrogenation reaction is the ability to use solar heat to increase the electrolyte temperature and the resulting kinetics of the homogeneous hydrogenation catalyst. We have highlighted the thermal advantage of our coupled concept on page 14 of our manuscript.

Associated changes to the manuscript:

- Figure R3 is added to the Supporting Information as Figure S12.
- Page 15: “More generally, the low conversion at high current densities indicates a mismatch in the rates of HER with that of the hydrogenation. **Indeed, H₂-to-MSA conversion decreases with increasing current density values (Fig. S12).**”
- Page 15: “This illustrates the benefit of the efficient thermal coupling between the light absorber and the catalyst that is inherent to PEC devices (and much more difficult to achieve in wired photovoltaic-electrolyzer devices **or in systems where the H₂ generation and hydrogenation are performed in separate reactors**).”

6. As the work of Holzhäuser et al. in 2017 was mentioned, why the authors used Pt rather than Pb? I wonder if the direct and Rh/TPPTS-catalyzed IA hydrogenation can be coupled on Pb system.

Response: We thank the reviewer for the comment. Pt was selected in our study since it can electrochemically generate H₂ efficiently based on its favorable binding energy of hydrogen. On the other hand, hydrogen weakly adsorbs on Pb requiring large thermodynamic overpotential to form Pb-H intermediates. Direct and Rh/TPPTS-catalyzed hydrogenation using a Pb electrode would theoretically be possible; in this case, the H₂ byproduct would be converted homogeneously to MSA leading to selectivity higher than that reported by Holzhäuser et al. in 2017. However, as already mentioned in the main text of our manuscript, the onset potential of this system was reported to be -1 V vs. RHE. This overpotential of 1 V is rather huge (comparable to the thermodynamic potential of water splitting, 1.23 V), and it is not clear whether the homogeneous catalyst remains stable and not electrochemically reduced within this potential range. For these reasons, we keep the use of Pt in our device and consider the system proposed by the reviewer to be beyond the scope of this study.

Associated changes to the manuscript:

- Page 9: “**The MSA yield of direct hydrogenation using Pb might be improved by adding Rh/TPPTS catalyst; Rh/TPPTS could utilize the H₂ generated as a byproduct at the Pb electrode to hydrogenate IA to MSA. However, considering the huge overpotential on Pb and that the stability of Rh/TPPTS within the potential range is unknown, we keep this concept of combining direct and coupled hydrogenation in a single device for future investigations.**”

7. In Fig. 4, it is difficult to recycle the homogeneous catalysts in Feedstock A and B. The separation of catalysts and products from the obtained mixture may be costly and challenging.

Response: We appreciate and share the same concern of the reviewer regarding the cost and challenges associated with separating the products and catalysts. We have indeed included the product separation cost factor in our techno-economic assessment. The consideration was already included (see Table S4), and we have now included the references that justify our assumptions. From the cost breakdown example shown in Table S5, the separation cost is indeed not insignificant (~17% of the total system cost). However, this factor has been considered in evaluating the economic merit of solar hydrogenation vs. solar water splitting, which we discussed and showed in the manuscript and Table 1.

Associated changes to the manuscript:

- Table S4 has been modified to include references that justify the assumptions taken.

Reviewer #2 (Remarks to the Author):

This article reports on the in situ use of (photo)electrochemically generated H₂ for the homogeneous hydrogenation of itaconic acid to methyl succinic acid. ~60% of the produced hydrogen can be utilized under optimized condition. Additionally, a techno-economic assessment of the proposed device further indicated the benefit of coupling solar hydrogen production to a chemical transformation. This makes full use of the cathode in the solar electrochemical cell. The idea is novel and the article is well organized. It is suitable for publication in NC. The following point might be considered before publication.

1. More description about the catalytic mechanism of Rh/TPPTS catalyst in the system is needed.

Response: We thank the reviewer for the suggestion. The catalytic mechanism of homogeneous hydrogenation using Rh/TPPTS has been reported in various publications.^{R1} We have now added a sentence in our manuscript that refers to these publications.

Associated changes to the manuscript:

- Page 4: “*The mechanism of homogeneous hydrogenation using a Rh/TPPTS catalyst has been reported elsewhere.¹⁵⁻¹⁷*”

2. Direct electrochemical IA hydrogenation to MSA demonstrated by Holzhäuser has low stability. How about the direct electrochemical IA hydrogenation with Rh/TPPTS catalyst at low current density (such as 2 mA/cm²)?

Response: We appreciate the comment from the reviewer. As also mentioned in our response to comment #6 of Reviewer 1, direct and Rh/TPPTS-catalyzed hydrogenation using a Pb electrode would be possible, and the byproduct, H₂, would be converted to MSA leading to selectivity higher than that reported by Holzhäuser et al. in 2017. However, the stability of the Rh/TPPTS at the working potential of the Pb electrode is unknown, and we suspect that the direct electrochemical hydrogenation pathway would be deactivated leading to higher overpotential at a given current density. In addition, as we mentioned in the main text, the onset potential was reported to be -1 V vs. RHE on Pb. This 1 V overpotential is significant, which is comparable to the thermodynamic voltage of water splitting, 1.23 V.

Associated changes to the manuscript:

- Page 9: “*The MSA yield of direct hydrogenation using Pb might be improved by adding Rh/TPPTS catalyst; Rh/TPPTS could utilize the H₂ generated as a byproduct at the Pb electrode to hydrogenate IA to MSA. However, considering the huge overpotential on Pb and that the stability of Rh/TPPTS within the potential range is unknown, we keep this concept of combining direct and coupled hydrogenation in a single device for future investigations.*”

3. It is predicted that the use Pt as a model cathode in this study can be replaced with alternative earth-

^{R1} Osborn, John A., et al. *Journal of the Chemical Society A: Inorganic, Physical, Theoretical* (1966): 1711-1732; Tanchoux, Nathalie, and Claude de Bellefon. *European Journal of Inorganic Chemistry* 2000.7 (2000): 1495-1502; Milano-Brusco, Juan S., et al. *Industrial & Engineering Chemistry Research* 47.20 (2008): 7586-7592.

abundant HER catalysts such as NiMo, transition metal dichalcogenides, and phosphides. It would be great to provide some preliminary results about it?

Response: We appreciate the suggestion of the reviewer. One consideration is related to the interaction between the homogeneous catalysts, excess ligands, and/or organic substrates with the electrode surface, which may affect the efficiency and stability. We have shown in this study that this consideration is not an issue for Pt, which is why it was used in our demonstration devices. At this point, it is not clear whether this is also the case for the alternative earth-abundant HER catalysts. Indeed, we are working on this right now, but the results are far too premature to be reported at this stage. We kindly ask for the understanding of the reviewer that these results are therefore beyond the scope of the current study.

4. Does the potential of cathode affect the utilization of in-situ produced hydrogen?

Response: We thank the reviewer for the question, and we kindly refer to our response to comment #4 of Reviewer 1.

5. How about the selectivity in the reaction of IA hydrogenation to MSA for this system?

Response: As shown in Table S1, the total Faradaic efficiency of our device is very close to 100%. NMR spectra also do not show any signals other than those of IA and MSA. We have therefore confirmed that there is no side reaction in the catholyte chamber other than the H₂ generation from the cathode and the coupled hydrogenation of IA to MSA. This has been clearly discussed in our manuscript, as reproduced in this response letter below.

Page 15: *“Direct detection of H₂ using a mass spectrometer has also been conducted using an electrochemical cell (in the dark) with a Pt cathode at operating currents that are similar to those of the coupled PEC and the PV-EC hydrogenation devices above. As shown in Table S1, the H₂ collection efficiencies at -3 and -50 mA are 34% and 81%, respectively. Adding these values to the respective H₂-to-MSA conversion efficiencies results in total Faradaic efficiency values close to 100%. In other words, there is no side reaction in the catholyte chamber other than the H₂ generation from the cathode and the subsequent hydrogenation of IA to MSA. The 100% total Faradaic efficiency also suggests that the presence of Rh/TPPTS does not impact the effectiveness of H₂ generation at the Pt cathode.”*

6. In the techno-economic assessment, it would be great to include the technology of PV-EC combined with traditional hydrogenation to highlight the (photo)electrochemically generated H₂ for the homogeneous hydrogenation.

Response: We thank the reviewer for the suggestion. We currently have not included the comparison of our coupled PEC hydrogenation device with the combination of PV-electrolysis and conventional hydrogenation. Doing so, as the reviewer suggested, would of course be great, but this unfortunately requires a rather significant effort in building the complete life cycle inventory and cost of the PV-electrolysis + conventional hydrogenation. Since this comparison does not necessarily affect the main messages of our current manuscript, we hope the reviewer kindly understands that this is beyond the scope of our current study. Nevertheless, in our previous life-cycle net energy assessment study,^{R2} we have calculated the cumulative energy demand (CED) of conventional MSA generated through hydrogenation

^{R2} Zhang et al. *Nat. Commun.* 14, 2023, 991

using H₂ from steam methane reforming (SMR) to be 84 MJ/kg_{MSA}. A similar CED value of MSA is expected when the H₂ is replaced from that obtained using PV-electrolysis; this is because the CED values of H₂ from SMR and H₂ from PV-electrolysis are similar (183.2 vs. 187.5 MJ/kg_{H₂} – see Table S5 in ref. R2). In contrast, the CED of MSA using our coupled PEC device (base case condition of 5% STH efficiency and 10-year device lifetime) was calculated to be 13.3 MJ/kg_{MSA}. Since a lower energy demand typically translates to a reduced cost, one can expect that our coupled PEC device is also more economically competitive than combining PV-electrolysis with conventional hydrogenation. We have added brief sentences to our current manuscript to highlight this point.

Associated changes to the manuscript:

- Page 16: *“Very recently, we have performed and reported a life cycle analysis study in which it is shown that introducing the coupled hydrogenation of IA to MSA to a photoelectrochemical hydrogen production device significantly improves the net energy balance and decreases the energy payback time of the overall device.³⁶ In that study, we also showed that the MSA obtained from our coupled photoelectrochemical hydrogenation device requires much lower cumulative energy demand than that obtained from conventional hydrogenation using hydrogen from steam methane reforming or PV-electrolysis.”*

Reviewer #3 (Remarks to the Author):

The manuscript of Abdi and coworkers entitled “Solar-drive upgrading of biomass by coupled hydrogenation using in situ (photo)electrochemically generated H₂” proposes the use of the H₂ generated either electrochemically or photoelectrochemically to turn itaconic acid into methyl succinic acid using a homogeneous catalyst dissolved in the electrolyte. The manuscript is interesting, in the sense that it opens a new gateway to utilize PEC devices in a more economically-competitive fashion. Indeed, it is a follow up paper from a previous article these authors published in this same journal (DOI: 10.1038/s41467-023-36574-1), cited as REF. 32. While in that previous report the authors mainly proposed the concept of the “coupled hydrogenation” and the economic viability of this “indirect” hydrogenation approach, here, the authors describe the experimental evidence and elaborate on the aspects that control the performance. Overall, the results and conclusions align well with the requirements for publication in nature communications, and therefore I would support the acceptance after completing a minor revision to clarify some aspects:

1. In page 7 (Line141), the authors claim that “cathodic shoulder” is indicative of the direct heterogeneous hydrogenation. What are the authors referring to? The authors should provide some experimental evidence that supports their claim on the “deactivation” of the heterogeneous reaction. If not, it seems quite hasty to rule out the participation of such a reaction.

Response: We thank the reviewer for bringing up this point. We refer to the cathodic feature in the range between 0 and -0.05 V vs. RHE (Fig. 1a), which is only present when IA is present in the catholyte but not the Rh/TPPTS catalyst. We argue that this corresponds to the direct heterogeneous hydrogenation pathway on the surface of the Pt cathode; consequently, the absence of this feature when both IA and Rh/TPPTS catalysts are present in the catholyte can be interpreted by the absence of the direct heterogeneous hydrogenation pathway. Note that the cathodic feature is also absent when no hydrogenation reaction occurs in the catholyte that does not contain IA and Rh/TPPTS. Nevertheless, we agree with the reviewer that our

claim remains a proposed explanation. We, therefore, have modified the sentences in the main text to address this concern.

Associated changes to the manuscript:

- Page 7: “Interestingly, although the cathodic shoulder that we attributed to the occurrence of direct heterogeneous IA hydrogenation (between 0 and -0.05 V vs. RHE) disappears, MSA is clearly detected from the product analysis (Fig. 1b bottom – blue). This can only be explained by homogeneous IA hydrogenation over the Rh/TPPTS catalyst, using the hydrogen that was generated in situ at the Pt surface.”

2. The authors conclude that the rate of H₂ production critically affects the H₂ concentration in the electrolyte and hence the conversion rate of the itaconic acid. While the authors explored the effect of the catholyte flow and the H₂ bubble size, did the authors evaluate the effect of the current density on the selectivity? It would be interesting to show how the selectivity changes as a function of the current density.

Response: We thank the reviewer for the suggestion. Please refer to our response to comment #4 of Reviewer 1.

3. The LCOH analysis provided by the authors did not consider the separation process of the product. While they propose some techniques this was not explicitly contemplated in the LCOH. Could the authors estimate the cost for the separation process to provide a more complete picture of the whole process?

Response: We would like to clarify that our LCOH analysis has actually already considered the product separation cost. We refer to our process flow diagram in Fig. S17, which is reproduced as Fig. R4 in this response letter, in which the product separation and gas handling components are included within our system boundary. Table S4 provides the assumptions considered for our coupled photoelectrochemical hydrogenation device, and the balance-of-system cost for the separation unit and the O&M cost for MSA separation were already included. We have now included the references that justify our assumptions. From the cost breakdown example shown in Table S5, the cost related to the separation can be seen, which accounts for ~17% of the total system cost.

Figure R4. Simplified process flow diagram for the techno-economic analysis of our coupled photoelectrochemical hydrogenation device. The dashed red line indicates the system boundary or scope of our study (cradle-to-gate), from raw material extraction to product separation and the end-of-life treatment of the device.

Associated changes to the manuscript:

- Table S4 has been modified to include references that justify the assumptions taken.
- The caption of Fig. S17 has been modified to emphasize the inclusion of product separation in our techno-economic assessment: “**Figure S17.** *Simplified process flow diagram for the techno-economic analysis of our coupled photoelectrochemical hydrogenation device. The dashed red line indicates the system boundary or scope of this study (cradle-to-gate), from raw material extraction to **product separation and** the end-of-life treatment of the device.*”

REVIEWER COMMENTS

Reviewer #1 (Remarks to the Author):

Although the author has responded to the comments point by point, I think there are still some issues that need to be addressed.

1. Although the author responded that they were not aware of any reports of coupled photoelectrochemical production of hydrogen with homogeneous hydrogenation, I noticed that "coupling the photoelectrochemical production of hydrogen with the Rh/TPPTS-catalyzed hydrogenation of itaconic acid (IA) to methyl succinic acid (MSA)" has been reported as Nat. Commun. 2023, 14, 991. Moreover, the sentence "We note that this optimistic scenario is realistic since our preliminary experiments have demonstrated an H₂-to-MSA conversion efficiency of up to 60%" suggests that the mechanism of this efficient H₂-to-MSA conversion has been revealed in Nat. Commun. 2023, 14, 991. The author should carefully consider the innovation that distinguishes this manuscript from Nat. Commun. 2023, 14, 991.
2. "Rh/TPPTS can prevent the direct electrochemical IA hydrogenation on Pt" indicates that the Rh/TPPTS influences the reaction selectivity on Pt (as the response of the author), and I also wonder if Rh/TPPTS influences the reaction rate of H₂ production on Pt.
3. I think the fitting curve of the relationship between the HER rate (current density) and the H₂-to-MSA conversion should theoretically pass through the origin, but Figure R3 merely indicates that the stronger HER the lower H₂-to-MSA conversion. I wonder if there is an optimum in the range of 0-0.3 mA/cm².
4. According to the response to comment #5, I still cannot agree with the necessity of integrating the photoelectrochemical HER and the hydrogenation reaction. One reason is the hydrogenation reaction can also be heated by sunlight in a separated reactor, while this hydrogenation reaction can be further enhanced (for instance, adding some photothermal catalysts).
5. According to the response to comment #5, the author stated that the hydrogenation reaction occurs homogeneously in the catholyte solution, so that the applied bias will not directly influence the hydrogenation reaction. However, in the response to comment #6, the author refused to consider Pb as the electrode on the grounds that it is not clear whether the homogeneous catalyst remains stable and not electrochemically reduced within this potential range. I think these responses are contradictory to each other.

Reviewer #2 (Remarks to the Author):

All the comments have been well addressed, the revised manuscript is ready for publication.

Reviewer #3 (Remarks to the Author):

After carefully reviewing the manuscript and the comments I consider that the authors have addressed all the questions and the manuscript does not need further revision.

Response to reviewers' comments

Manuscript ID: NCOMMS-23-13885-A

Title: Solar-driven upgrading of biomass by coupled hydrogenation using in situ (photo)electrochemically generated H₂

In this response letter, the reviewers' comments to our original manuscript are provided in **black**, and our point-by-point responses as well as the corresponding changes to the manuscript are shown in **blue**.

Reviewer #1 (Remarks to the Author):

Although the author has responded to the comments point by point, I think there are still some issues that need to be addressed.

1. Although the author responded that they were not aware of any reports of coupled photoelectrochemical production of hydrogen with homogeneous hydrogenation, I noticed that “coupling the photoelectrochemical production of hydrogen with the Rh/TPPTS-catalyzed hydrogenation of itaconic acid (IA) to methyl succinic acid (MSA)” has been reported as Nat. Commun. 2023, 14, 991. Moreover, the sentence “We note that this optimistic scenario is realistic since our preliminary experiments have demonstrated an H₂-to-MSA conversion efficiency of up to 60%” suggests that the mechanism of this efficient H₂-to-MSA conversion has been revealed in Nat. Commun. 2023, 14, 991. The author should carefully consider the innovation that distinguishes this manuscript from Nat. Commun. 2023, 14, 991.

Response: We thank the reviewer for the careful evaluation of our work. We need to point out that our previously published paper (Nat. Commun. 2023, 14, 991) only reported a life-cycle net energy balance assessment of the coupled photoelectrochemical hydrogen production and hydrogenation concept; there was no experimental work or report in that paper. The statement cited by the reviewer referred to preliminary experiments that we are reporting on *now*, for the first time, in the *current* manuscript. Therefore, the reviewer's comment that the mechanism of the efficient H₂-to-MSA conversion has been revealed in our previous paper is a misunderstanding. We emphasize that all experimental work regarding the mechanism as well as demonstration of the concept are reported for the first time in the current manuscript. We outline a few innovation points that clearly distinguish our current manuscript from the recently published one:

- The concept of coupled photoelectrochemical hydrogen production with homogenous hydrogenation is experimentally demonstrated for the first time.
- We showed experimentally that the stability of the coupled (photo)electrochemical hydrogenation system is superior to the direct electrochemical hydrogenation system.
- We performed multiphysics simulations and combined kinetic analysis of the coupled (photo)electrochemical hydrogenation reaction.
- Technoeconomic assessments of the coupled photoelectrochemical hydrogenation concept are reported for the first time (thus complementing the life cycle / net energy balance assessment of the previous paper).

We believe these innovation points justify the need and impact of publishing our current manuscript in Nature Communications. These points have been explicitly conveyed in the Abstract (see below –

highlighted with underlines). We have also modified our Conclusion section to contain these points more explicitly (see below – highlighted in yellow).

Abstract

With the increasing pressure to decarbonize our society, green hydrogen has been identified as a key element in a future fossil fuel-free energy infrastructure. Solar water splitting through photoelectrochemical (PEC) approaches is an elegant way to produce green hydrogen, but for low-value products like hydrogen, PEC production pathways are difficult to be made economically competitive. A possible solution is to co-produce value-added chemicals. Here, we propose and demonstrate the in situ use of (photo)electrochemically generated H₂ for the homogeneous hydrogenation of itaconic acid—a biomass-derived feedstock—to methyl succinic acid. Coupling these two processes offers major advantages in terms of stability and reaction flexibility compared to direct electrochemical hydrogenation, while minimizing the overpotential. An overall conversion of up to ~60% of the produced hydrogen has been demonstrated for our coupled process, and a techno-economic assessment of our proposed device further reveals the benefit of coupling solar hydrogen production to a chemical transformation.

Conclusions

In summary, we have demonstrated the concept of solar-driven upgrading of biomass-derived feedstocks by in situ utilization of (photo)electrochemically generated H₂ to perform a hydrogenation reaction over homogeneous catalysts dispersed in the catholyte. By taking the hydrogenation of IA to MSA as a model reaction and Rh-TPPTS as the homogenous catalyst, we have shown that the coupled process is more stable and far less sensitive to deactivation than the direct heterogeneous electrochemical hydrogenation. Through a combination of multiphysics simulations and experiments, we found that the catalytic reaction rate and bubble management are critical parameters in controlling hydrogenation and product conversion. A H₂-to-MSA conversion of up to 60% was achieved with coupled electrochemical hydrogenation (dark), and solar-driven coupled hydrogenation devices based on PEC and PV-EC approaches were successfully demonstrated with a H₂-to-MSA conversion of up to 53%. Using a photoanode leads to higher conversion than using an electrolyzer since the reaction rate of the former matches better with the hydrogenation rate. A techno-economic analysis shows that by coupling the PEC water splitting reaction with the in situ catalytic hydrogenation of IA, the levelized cost of hydrogen could be dramatically reduced to the point that it becomes competitive even with generating hydrogen from steam methane reforming. Combined with the favorable life-cycle net energy balance we recently reported,³⁶ these results clearly demonstrate the overall benefit of coupling a hydrogenation reaction with a PEC H₂ production. Finally, the coupled PEC/catalysis approach can be used for a wide variety of chemical transformations through the appropriate design of homogeneous catalysts and offers a promising pathway to make solar hydrogen production economically feasible.

2. “Rh/TPPTS can prevent the direct electrochemical IA hydrogenation on Pt” indicates that the Rh/TPPTS influences the reaction selectivity on Pt (as the response of the author), and I also wonder if Rh/TPPTS influences the reaction rate of H₂ production on Pt.

Response: We appreciate the reviewer's comment related to the reaction rate in the presence of Rh/TPPTS homogenous catalysts. Indeed, we have shown in Figure S6 (reproduced below as Figure R1a for

convenience) that the potential needed to sustain a -2 mA cm^{-2} current density becomes more negative in the presence of Rh/TPPTS catalyst (see magenta curve vs. black curve for pure KP_i). This need for a stronger driving force for electrochemical reduction implies the passivation of highly active HER sites on the Pt surface, such as the (110) and (100) facets, as already discussed in Supplementary Note S2 and Figure S3. Therefore, as the reviewer commented, the HER rate indeed decreases in the presence of Rh/TPPTS catalyst. We have now confirmed this by performing linear sweep voltammetry measurements in 1 M KP_i , with and without the Rh/TPPTS catalyst. As shown in Figure R1b, the presence of the Rh/TPPTS catalyst increases the overpotential and decreases the magnitude of the current density. We have now added this data and a brief discussion to our revised manuscript and supplementary information.

Figure R1. (a) Chronopotentiometry at -2 mA cm^{-2} with various additives in the electrolyte (reproduced from Figure S6). (b) Linear sweep voltammograms of a Pt cathode in 1 M KP_i electrolyte, with and without the Rh/TPPTS catalyst.

Associated changes to the manuscript:

- Figure S6 has been updated with Figure R1.
- The caption of Figure S6 has been modified as follows: “**Figure S6.** (a) Chronopotentiometry at -2 mA cm^{-2} with various additives in the electrolyte. (b) Linear sweep voltammograms of a Pt cathode in 1 M KP_i , with and without Rh/TPPTS catalyst. The Pt electrode shows stable H_2 production in 1 M KP_i , as shown by the black curve. When IA is added to the electrolyte without any Rh/TPPTS catalyst (red and green curves), a gradual decrease of the measured potential is observed indicating that the heterogenous hydrogenation is deactivated during the course of the cathodic reactions. Only in the presence of Rh/TPPTS catalyst (blue and magenta curves) are the cathodic potentials stable suggesting the prevention of the heterogeneous hydrogenation reaction and the suppression of the deactivation of the Pt electrode. It is also noted that the HER overpotential without IA (i.e., in pure KP_i) increases in the presence of Rh/TPPTS catalyst (magenta curve vs. black curve), which implies the passivation of highly active HER sites on the Pt surface, such as the (110) and (100) facets, which are also adsorption sites of IA as discussed in Figure S3.”

- Page 7: “Control measurements with various additives suggest that adding the Rh catalyst is key to prevent deactivation associated with the presence of IA in the catholyte (Fig. S6a), despite the slight increase of overpotential (Fig. S6b).”

3. I think the fitting curve of the relationship between the HER rate (current density) and the H₂-to-MSA conversion should theoretically pass through the origin, but Figure R3 merely indicates that the stronger HER the lower H₂-to-MSA conversion. I wonder if there is an optimum in the range of 0-0.3 mA/cm².

Response: We first clarify that the H₂-to-MSA conversion should *not* theoretically pass through the origin. The definition of H₂-to-MSA conversion ($\eta_{\text{H}_2\text{-to-MSA}}$), as also shown in equation 3 of the manuscript, is the following:

$$\eta_{\text{H}_2\text{-to-MSA}} = \frac{r_{\text{MSA}}}{j_{\text{app}}A/2F} \quad (\text{R1})$$

where r_{MSA} , j_{app} , A , and F are the measured MSA production rate, the applied current density, the electrode area, and the Faraday constant, respectively. At the origin point of the plot (i.e., zero current density), the denominator of equation R1 above is zero, and H₂-to-MSA conversion can therefore not be defined.

Regarding the shape of the current density vs. H₂-to-MSA conversion curve, an optimum is actually not expected at lower current density range. At low current density, the produced H₂ would mainly remain as dissolved H₂. The H₂-to-MSA conversion is therefore dominated by the homogenous reaction rate and the residence time, as discussed in Figure 2. Therefore, we expect that the H₂-to-MSA conversion would saturate at low enough current density (close to 100%). At high current density, the produced H₂ would mainly evolve as bubbles, leaving a much smaller fraction of H₂ that can dissolve. The H₂-to-MSA conversion is therefore expected to decrease. In other words, an inverted S curve is expected when plotting the current density vs. H₂-to-MSA conversion. To confirm this, we have now extended the current density range in our experiments to 0.1 – 10 mA cm⁻² (three orders of magnitude). As shown in Figure R2 below, H₂-to-MSA conversion values start to saturate at ~75% below 1 mA cm⁻² and ~5% above 5 mA cm⁻². This agrees very well with our expectation of the inverted S curve and confirms that there is no optimum in the range of 0 – 0.3 mA cm⁻² that the reviewer was wondering about.

Figure R2. H₂-to-MSA conversion values obtained from a Pt electrode (in dark conditions) as a function of the current density. The electrolyte was 1 M KPi with 0.15 M itaconic acid (IA) and 0.9 mM Rh/TPPTS catalyst.

Associated changes to the manuscript:

- Figure S14 (previously Figure S13) has been updated with Figure R2.
- Page 15: “Indeed, H₂-to-MSA conversion decreases with increasing current density values (Fig. S14). At low enough current density, the H₂-to-MSA conversion saturates since the generated H₂ mainly remains as dissolved H₂ and the conversion is determined by the residence time and the homogenous hydrogenation rate, as discussed in Figure 2. This suggests that photoelectrochemical devices, which typically show current densities 25-100× lower than those for electrolyzers, are particularly well-suited for coupling with homogeneous hydrogenation reactions.”

4. According to the response to comment #5, I still cannot agree with the necessity of integrating the photoelectrochemical HER and the hydrogenation reaction. One reason is the hydrogenation reaction can also be heated by sunlight in a separated reactor, while this hydrogenation reaction can be further enhanced (for instance, adding some photothermal catalysts).

Response: We agree with the reviewer that separating the PEC H₂ evolution and the hydrogenation reaction is also possible. We in fact showed this by demonstrating catalytic hydrogenation of IA to MSA using a separate H₂ feed. The reviewer is also right to point out that the solar thermal integration advantage remains speculative; we believe it is an important argument, but it has yet to be quantitatively demonstrated. The reviewer’s suggestion regarding the use of solar heating in a separate reactor is also possible. However, it is similarly unclear at this point, whether the need for further land area to collect the sunlight to heat the reactor can be justified.

Therefore, we present a few important arguments highlighting the benefit of integrating PEC H₂ evolution and hydrogenation vs. performing them sequentially in separate reactors. First, performing the

hydrogenation reaction in an integrated system will benefit from the large interface between the photoelectrochemically generated H₂ over a large area (i.e., the photoelectrode's surface) and the liquid electrolyte. This large interface promotes a high degree of interaction between H₂ and the hydrogenation catalyst and substrate in the electrolyte. In non-integrated systems, H₂ would be collected and then fed back to a reactor in a gaseous form. Additional steps would be needed to ensure they are distributed homogeneously inside the hydrogenation reactor. Indeed, it is well-known that it can take a lot of effort to increase the interface area between H₂ and the liquid in a hydrogenation tank (e.g., stirring vigorously, the addition of a gasification membrane), which can be avoided in an integrated system. Second, in the case of IA to MSA, we have discussed in our response to previous comment #6 of Reviewer 2 that the cumulative energy demand (CED) of MSA using PV-electrolysis + conventional hydrogenation is much higher (~6-fold) than the CED of MSA using our coupled PEC device. Since a lower energy demand typically translates to a reduced cost, our integrated system is also expected to be economically more competitive than the non-integrated system. Finally, the advantage of separating the two processes would be that each process can be individually optimized, yielding a higher overall performance. However, for a homogeneous reaction such as the proposed hydrogenation reaction, the optimization parameters are the concentration of catalysts, feedstock/substrate, and the flow rate. All these parameters can also be conveniently adjusted in the catholyte compartment of our integrated coupled PEC system. We have now added this discussion to the revised manuscript.

Associated changes to the manuscript:

- Page 18: *"It is noted that the PEC H₂ generation and the hydrogenation of IA to MSA can also be performed sequentially in separate reactors, i.e., in a non-integrated fashion. However, our integrated coupled PEC H₂ generation and hydrogenation system offers several advantages. Most importantly, performing the hydrogenation reaction in an integrated system will benefit from the large interfacial area between the photoelectrode surface (where the H₂ is generated) and the liquid electrolyte (where the hydrogenation takes place). This ensures short diffusion distances and optimal interaction between the reactants and catalysts. In non-integrated systems, significant engineering efforts would be needed to ensure homogeneous mixing and optimal interaction between the separately supplied H₂ and the IA + catalyst already present in the solution (e.g., by stirring vigorously, or by adding a gasification membrane); these steps can be avoided in an integrated system. One argument for separating the two processes would be that each can be individually optimized, yielding higher overall performance. However, for a homogeneous reaction such as the hydrogenation of IA to MSA, all of the optimization parameters (i.e., the concentration of catalysts, feedstock/substrate, and the flow rate) can also be conveniently adjusted in the catholyte compartment of our integrated coupled PEC system. Another potential advantage of an integrated system is that the solar heating of the photoelectrode may benefit the catalytic reaction rates, although the magnitude of this effect still needs to be quantified."*

5. According to the response to comment #5, the author stated that the hydrogenation reaction occurs homogeneously in the catholyte solution, so that the applied bias will not directly influence the hydrogenation reaction. However, in the response to comment #6, the author refused to consider Pb as the electrode on the grounds that it is not clear whether the homogeneous catalyst remains stable and not electrochemically reduced within this potential range. I think these responses are contradictory to each other.

Response: We apologize for the misunderstanding. Our responses are not contradictory to each other given the context of each statement. Our claim that the applied bias does not influence the hydrogenation (other than the indirect effect on current density, i.e., our response to the previous comment #5) is valid within our experimental conditions using a Pt electrode. Of course, this statement needs to be evaluated again for a much larger potential window if a Pb electrode is used instead (as we mentioned in our previous rebuttal, HER on Pb doesn't start until about -1 V vs RHE, indicating a huge overpotential). We also would like to clarify that we did not categorically refuse to consider Pb, but we argue in our response to the previous comment #6 that it is beyond the scope of our manuscript as more investigations—which do not affect the main messages of our study—are needed.

In addition, we have now experimentally determined the stability of the Rh/TPPTS catalyst. It has been reported that instability of Rh/TPPTS would be accompanied by a color change of the solution.^[R1-R3] We, therefore, measured the UV-vis absorption spectra of the catholytes at various stages: as-prepared (i.e., fresh) and after applying various current densities. As shown in Figure R3, the spectra practically overlap with each other, confirming the stability of the Rh/TPPTS within our experimental conditions using a Pt electrode. We have now added this data and a brief discussion to the revised manuscript and supplementary information.

R1. T. Schmidt et al. *Adv. Synth. Catal.* 351 (2009) 750-754

R2. M. Schmidt et al. *Ind. Eng. Chem. Res.* 58 (2019) 2445-2453

R3. N. Zhang et al. *J. Org. Chem. Res.* 4 (2016) 13-22

Figure R3. UV-vis absorbance (1 – Transmission) of the catholyte (1 M KP_i + 0.15 M IA + 0.9 mM Rh/TPPTS catalyst) at various stages: as-prepared (i.e., fresh) and after exposure to various current densities for 1 hour each.

Associated changes to the manuscript:

- Figure R3 has been added to the supplementary information as Figure S7.

- Page 7/8: “*The Rh catalyst is also stable within our experimental conditions using the Pt electrode (Fig. S7).*”
- Page 27: “*UV-vis transmission spectroscopy was performed using a white light source (deuterium-halogen lamp, DH-2000-BAL, Ocean Optics) and a CCD spectrometer (Maya 2000-Pro, Ocean Optics) coupled with optical fibers and collimators.*”

Concluding response to Reviewer #1 comments

We thank the reviewer again for the extensive and careful evaluation of our manuscript. We believe we have addressed all the comments by the additional experiments and discussion, and we note that the main conclusion and messages of the manuscript are not affected by these changes. We hope the reviewer now finds our revised manuscript to be suitable for publication in Nature Communications.

Reviewer #2 (Remarks to the Author):

All the comments have been well addressed, the revised manuscript is ready for publication.

Response: We thank the reviewer for their efforts in evaluating our manuscript, and we are happy that the reviewer considered our manuscript to be ready for publication.

Reviewer #3 (Remarks to the Author):

After carefully reviewing the manuscript and the comments I consider that the authors have addressed all the questions and the manuscript does not need further revision.

Response: We thank the reviewer for the careful review of our manuscript, and we are pleased to know that we have satisfactorily addressed all the reviewer's concerns.